# Development and validation of the facial scale (FaceSed) to evaluate sedation in horses

**Alice Rodrigues de Oliveira[1]☯, Miguel Gozalo-Marcilla[2]☯, Simone Katja Ringer[3], Stijn Schauvliege[4], Mariana Werneck Fonseca[1]☯, Pedro Henrique Esteves Trindade[1]☯, José Nicolau Prospero Puoli Filho[1], Stelio Pacca Loureiro Luna[1]☯ \***

**1** Department of Veterinary Surgery and Animal Reproduction, School of Veterinary Medicine and Animal Science, São Paulo State University (UNESP), Botucatu, São Paulo, Brazil, **2** The Royal (Dick) School of Veterinary Studies and The Roslin Institute, The University of Edinburgh, Midlothian, United Kingdom, **3** Department of Clinical Diagnostics and Services, Section Anaesthesiology, Vetsuisse Faculty, University of Zürich, Zürich, Switzerland, **4** Department of Surgery and Anaesthesia of Domestic Animals, Faculty of Veterinary Medicine, Ghent University, Merelbeke, Belgium

☯ These authors contributed equally to this work.
\* stelio.pacca@unesp.br

**Data Availability Statement:** Data was included as a Supporting information file (Data.FaceSed).

**Funding:** São Paulo Research Foundation (FAPESP) for the doctoral scholarship granted

## Abstract

Although facial characteristics are used to estimate horse sedation, there are no studies measuring their reliability and validity. This randomised controlled, prospective, horizontal study aimed to validate a facial sedation scale for horses (FaceSed). Seven horses received detomidine infusion i.v. in low or high doses/rates alone (DL 2.5 µg/kg+6.25 µg/kg/h; DH 5 µg/kg+12.5 µg/kg/h) or combined with methadone (DLM and DHM, 0.2 mg/kg+0.05 mg/kg/h) for 120 min, or acepromazine *boli* i.v. in low (ACPL 0.02 mg/kg) or high doses (ACPH 0.09 mg/kg). Horses' faces were photographed at i) baseline, ii) peak, iii) intermediate, and iv) end of sedation. After randomisation of moments and treatments, photos were sent to four evaluators to assess the FaceSed items (ear position, orbital opening, relaxation of the lower and upper lip) twice, within a one-month interval. The intraclass correlation coefficient of intra- and interobserver reliability of FaceSed scores were good to very good (0.74–0.94) and moderate to very good (0.57–0.87), respectively. Criterion validity based on Spearman correlation between the FaceSed *versus* the numerical rating scale and head height above the ground were 0.92 and -0.75, respectively. All items and the FaceSed total score showed responsiveness (construct validity). According to the principal component analysis all FaceSed items had load factors >0.50 at the first dimension. The high internal consistency (Cronbach´s α = 0.83) indicated good intercorrelation among items. Item-total Spearman correlation was adequate (rho 0.3–0.73), indicating homogeneity of the scale. All items showed sensitivity (0.82–0.97) to detect sedation, however only orbital opening (0.79) and upper lip relaxation (0.82) were specific to detect absence of sedation. The limitations were that the facial expression was performed using photos, which do not represent the facial movement and the horses were docile, which may have reduced specificity. The FaceSed is a valid and reliable tool to assess tranquilisation and sedation in horses.

(ARO), protocol 2017/16208-0, thematic project (SPLL) 2017/12815-0 and post-doctoral grant (M. G.M) 2017/01425-6. Funder website: https://fapesp.br/ The funders had no role in study design, data collection and analysis, decision to publish, or preparation of the manuscript.

**Competing interests:** The authors have declared that no competing interests exist.

## Introduction

Sedation and tranquilisation for procedures in standing horses are alternatives to general anaesthesia [1] to reduce the high anaesthesia-related mortality (0.9%) in these species [2, 3]. Frequently used for premedication before general anaesthesia or longer procedures in standing position, the α-2 adrenergic receptor agonists are widely used and play an important role for chemical restraint in horses. They include xylazine, romifidine, detomidine, medetomidine, and dexmedetomine [4–8] and when combined with opioids, they produce a synergic analgesic effect [9], and minimise the opioid-induced excitement and intestinal hypomotility in horses [10, 11]. Acepromazine does not provide analgesia but is an option to provide mild to moderate tranquilisation in horses [10].

Different scales have been developed using scoring systems to qualify and quantify sedation in horses under the effect of sedative and tranquilisers [12]. The only objective measurement that does not require interpretation of the observer is head height above the ground (HHAG), first used in 1991 [13]. The HHAG is usually combined with other instruments to assess depth of sedation [12] and it is measured either by the distance in centimetres between the head and the floor [13] or the percentage of the head height compared to pre-treatment value [14]. Studies consider that an HHAG value below 50%, compared to the pre-treatment value, represents sufficient sedation [4, 5, 14]. The few divergences in the literature regarding the anatomical location for HHAG measurement; i.e., nostrils [14], chin [15, 16], or lower lips [17], may affect data reproducibility among different studies. Another limitation of the HHAG is that, although lowering the head is a typical and dose-dependent effect to assess $\alpha_2$ agonist-induced sedation, this method may not be applicable to assess the effects of other drugs, such as opioids and acepromazine.

Other methods to assess depth and quality of sedation are subjective because their interpretation is based on the experience of the observer with the effects of sedation in horses. They include the simple descriptive scale (SDS) [18], composite numerical rating scale (NRS) [19], and visual analogue scale (VAS) [20]. The visual analogue scale is represented by a line ranging from 0 (no sedation) to 10 cm (the most intense sedation possible) [1, 4, 14, 20–22]. It tends to correlate positively with the numerical rating scale comprised by the ordinal numbers from 0 to 10 [21, 23]. The composite numerical rating scale incorporates descriptions for different sedation intensities within each proposed item, in which the evaluator must assign one of the descriptions to each item [12, 13]. The simple descriptive scale consists of 0—no sedation, 1—mild, 2 –moderate, and 3—marked sedation, which should be chosen by the appraiser [12, 24]. Unidimensional scales (VAS, NRS and SDS) may be biased to the interpretation and experience of the evaluator, generating differences in results with doubtful representativeness when comparing studies [25].

The need for horse handling is another limitation of some instruments used to assess sedation. The HHAG, visual analogue scale, and numerical rating scale were adapted for both clinical [26, 27] and experimental purposes [11]. Postural instability (ataxia) is usually part of sedation scales [5], as well as threat response and movement of the head or ears in response to i. tactile [11], ii. visual, or iii. auditory stimuli [26]. These stimuli are i. touching the limb coronary band or inside the ears with a blunt object [8, 13, 19], ii. clapping hands [8, 13, 19], metallic sounds [5], blowing a horn or shaking a plastic gallon with stones [28, 29], and iii. shaking a towel or opening an umbrella in front of the horse [5, 8, 13, 19]. These stimuli may be cumbersome, and their delivery may vary according to the handler. The lack of a standard procedure can create confounding factors and, above all, these stimuli disturb the horse and may interfere with the depth of sedation, which could be a problem, notably in clinical situations.

Because facial assessment is based only on observation, the development of a scale based on facial expression would avoid the aforementioned limitations of sedation assessing instruments. The relevance of facial expressions together with behavioural characteristics of different species were brought up by Charles Darwin as a way of expressing emotional states [30]. Facial expression has been widely used to evaluate pain in veterinary medicine, especially in horses [31–33], which are prone to changing their facial expression under different circumstances [34]. Facial expression is not a novelty to assess sedation in horses either. It has already been incorporated in simple descriptive [35, 36] and composite numerical scales [26] for this purpose. The expressions involve ear tip distance [35, 37] or movement to stimulation [26, 36], reduced eye alertness [35] or aperture [26], lip aperture [35], atonic lower lip [36], and lip oedema [37]. However the validity of these expressions for identifying sedation has not yet been assessed [26].

Once developed, any instrument requires investigation to either confirm or not its value for measuring a construct. This may be accomplished by a validation process similar to those reported in other studies that validated behavioural pain scales in cats [38], horses [39], cattle [40], pigs [41], and sheep [42] and facial pain scales in cats [43], horses [31, 33, 44], and sheep [45]. For this, the repeatability and reproducibility (intra- and interobserver reliability, respectively) are evaluated as well as validation of the three 'Cs', referring to content, criteria, and construct; the last tests the responsiveness of the scale [46]. In addition, the item-total correlation and internal consistency identify the importance of each item of the scale to guarantee its homogeneity [47], and how much the items correlate to each other, respectively [46]. The sensitivity and specificity calculate the percentage of true positives (sedated horses) and negatives (not sedated horses), respectively, and the distribution of scores informs about the proportion of subitems of each category according to the intensity of sedation (i.e., subitems within each category representing the highest scores should be predominant in the deepest sedation status) [41].

Although a recent study validated a behavioural scale to assess sedation in horses [48], it requires handling of the horses. In face of the limitations of the current methods used to qualify and quantify sedation and tranquilisation in horses, the objective of this study was to develop and validate a facial scale (FaceSed) to quantify sedation in horses based on facial characteristics, assessed without horse manipulation.

The data of the present study were collected in two previous simultaneous studies. The first aimed to identify a protocol that provided antinociception without excessive sedation [11] and the second targeted development and validation of a sedation scale based on general behaviour [48]. The head heights above the ground transformed into percentages (HHAG%) were the only previously published data, which were also [11, 48] used in the present study, only for comparison to the new proposed instrument. All other data presented here are original of the current study. The horses were tranquilised and/or sedated with different doses/rates of detomidine alone or associated with methadone (Phase I), predicted to obtain a moderate/deep sedation score and two doses of acepromazine (Phase II) predicted to obtain low and high degree of tranquilisation scores. We hypothesised that the proposed scale, according to the statistical standards of content, criteria, and construct validities, inter- and intraobserver reliability, item-total correlation, internal consistency, sensitivity, and specificity, and the cut-off point determination adequately measures depth of sedation and tranquilisation through facial expression.

## Materials and methods

This study was approved by the Ethics Committee on the Use of Animals (CEUA) for research at the School of Veterinary Medicine and Animal Science, São Paulo State University

(UNESP), Botucatu, SP, Brazil, under protocol 2017/0051. The data of this study were collected during other previously published studies with different aims: pharmacokinetics [49, 50], anti-nociceptive effects [11], and development and validation of a behavioral scale for assessing sedation in horses [48] submitted to intravenous infusion of detomidine and methadone. The only duplicity of data presented in this study compared to the previous ones [11, 48] is HHAG % which were collected on-site.

Three geldings and four female quarter horses and appaloosa crossbred from the same herd (9–11 years, 372–450 kg) owned by the Edgardia experimental farm, from the São Paulo State University (UNESP), Botucatu Campus, Brazil, were enrolled in the study. The horses were kept on pasture and fed with hay and commercial feed once a day. All horses were healthy based on normal physical and laboratory tests (blood count and biochemistry: urea, alkaline phosphatase, alanine aminotransferase, and gamma glutamyl transferase). The day before the study the horses were collected and kept in covered facilities with water *ad libitum* and access to the outside area. Solid fasting was established for two hours before the experiment began. Interventions for each individual horse were always performed at a fixed weekday and time (morning or afternoon) with at least seven days between treatments.

The study was divided into two phases performed with the same horses, subjected to all treatments. The interval between Phases I and II was eleven months. Except for HHAG%, Phase I data are exclusive for the present study, but were collected as part of other simultaneously performed studies when the horses were sedated with constant rate infusions (CRIs) of detomidine alone or associated with methadone [11, 48–50]. The experimental number of horses was based on this previous simultaneous study [11], where sample size was estimated at n = 7, according to a pilot study based on HHAG and mechanical, thermal, and electrical noci-ceptive stimuli results [51] (α = 0.05, β = 0.80) [11]. Moreover, the sample size was corrobo-rated with articles of similar methodologies [9, 52]. In the second phase (Phase II), the same horses were subjected to two *boli* of acepromazine (see flowchart in S1 Appendix).

## Phase I

Phase I was conducted simultaneously with the previous studies [11, 48, 50]. Detomidine was chosen because it is the most commonly used α-2 adrenergic receptor agonist for continuous infusion, there is available pharmacokinetic published data, is used worldwide and because previous studies have demonstrated the effect of detomidine and methadone combination [9, 50, 52].

After weighing and directing the horses to the six-square metres experimental room, the left jugular vein area was clipped, asepsis performed and a 14-gauge catheter [G14 x 70 mm—Delta Med Srl, Italy] was inserted and fixed for drug administration. The horses were then placed in the restraining stocks inside the experimental stall. The HHAG, which is an objective parameter used for the original study [11], in a recently published behavioural sedation scale study [48], and in the present study (FaceSed validity), was measured on-site in cm without disturbing the horse, using a scale attached to the wall 1.5 m aside. For convenience, HHAG measurements were transformed into percentages, considering the baseline as 100% of the HHAG as in other reports [14]. The baseline HHAG was measured in centimetres when the horses were not sedated and positioned on the stocks, just before administration of the treat-ments. This value was considered as 100% of the HHAG and used for comparison against the other time-points. Afterwards, the face of the horse was photographed from the lateral and oblique craniocaudal positions to take images of the baseline moment. Subsequently, tactile, auditory, and visual stimuli were performed to evaluate the depth of sedation (data published previously [11]).

Once baseline measurements were taken, one of the following intravenous (i.v.) treatments (bolus + CRI for 120 min) was administered in a random manner (for the original crossover study) [11], by one of the evaluators (M.G.M) unaware of treatment. The treatments were previously numbered and randomised using a website [53] for each horse, and the sequence registered by another author responsible for the CRIs (M.W.F). The treatments were DL—low detomidine dose [Eqdomin, 10 mg/ml—Ourofino Saúde Animal, São Paulo, Brazil] (2.5 μg/kg followed by 6.25 μg/kg/h CRI), DH—high detomidine dose (5 μg/kg followed by 12.5 μg/kg/h CRI), DLM—low detomidine dose with methadone [Mytedom 10 mg/ml—Cristália Produtos Químicos e Farmacêuticos Ltda, São Paulo, Brazil] (2.5 μg/kg of detomidine + 0.2 mg/kg of methadone followed by detomidine 6.25 μg/kg/h + methadone 0.05 mg/kg/h CRIs), and DHM—high detomidine dose with methadone (5 μg/kg of detomidine + 0.2 mg/kg of methadone followed by detomidine 12.5 μg/kg/h + methadone 0.05 mg/kg/h CRIs). The 120-min CRIs were administered after the drug bolus using two syringe drivers [DigiPump SR8x—Digicare Biomedical Techology Inc, Florida, USA, and Pilot Anaesthesia—Fresenius Vial, Brezins, France], one for each drug. The horses were kept in the stocks for four hours for blood sampling for previously published studies performed simultaneously [11, 49, 50]. Apart from the baseline time-point, HHAG and sedation evaluations and photographic records were performed *on-site* 5, 15, 30, 60, 90, 120, 150, 180, 210, and 240 min after initial bolus administration of each treatment. These time-points were selected according to the previous studies from our group using a similar methodology [9, 52], and based on the judgment that the effect of detomidine and methadone would be abated 120 minutes after the end of infusion [49, 50]. For the present study only photographic records were analysed. HHAG was transformed in percentage from data already reported in a previous study performed simultaneously [11].

## Phase II

Phase II was performed for this study and for another recently published study that developed and validated a behavioural sedation scale in horses [48]. The only duplicity of data from that study is the HHAG%. Eleven months after the phase I, new physical and laboratory evaluations certified the good health of the same horses. An i.v. bolus of acepromazine [Acepran 1%—Vetnil Indústria e Comércio de Produtos Veterinários Ltda, Louveira, São Paulo, Brazil] was injected at low (ACPL 0.02 mg/kg) or high dose (ACPH 0.09 mg/kg) and the photos of the face of the horses were registered on-site at the time-points 5, 15, 30, 60, 90, 120, as indicated for phase I, but only up to 120 min, because during this phase there was no other concomitant study, therefore, unlike in Phase I, it was not necessary to maintain the horses in the stall for longer than 120 minutes, because they would be restless.

## Selection of photos and evaluations

The representative moments of sedation were selected for Phases I and II based on the reactions to tactile, auditory, and visual stimuli on-site [11], as the animals' faces were not evaluated on-site. For the present study, a total of 168 moments were selected for photographic evaluation (7 horses x 6 treatments x 4 representative moments). With the 7 horses and the 6 treatments described (four from Phase I and two from Phase II), the four representative moments of the different sedation intensities were the following: i) baseline, ii) peak sedation, iii) intermediate sedation and iv) end of sedation.

For Phase I, i) the baseline moment occurred before the administration of the drug(s), ii) the peak of sedation was considered 120 min after the bolus administration, immediately before the CRI(s) ended. This moment was selected according to the parallel pharmacokinetic study [49, 50], that showed high detomidine plasma concentrations, corroborated by the high

sedation scores obtained in the simultaneous parallel study, based on the tactile, auditory, and visual stimuli performed to evaluate the depth of sedation on-site and published elsewhere [11]. iii) Intermediate sedation occurred 30 min after peak sedation, 150 min after bolus administration (30 min after discontinuing drug infusion) and iv) the end of sedation was 240 min after bolus administration, characterised by low sedation scores according to the tactile, auditory, and visual stimuli recorded on-site, close to the baseline [11], and residual plasma drug concentrations [49].

For Phase II with *boli* of acepromazine, i) baseline moment occurred before the administration of the drug, ii) the peak of sedation was considered 60 min after the administration of acepromazine based on the results of the degree of sedation according to the tactile, auditory, and visual stimuli recorded on-site in the simultaneous parallel study [48] iii) Intermediate sedation occurred at 90 min when the score of tranquillisation registered on-site was moderate and iv) the end/reduction of tranquillisation when the on-site sedation score was low at 120 min after acepromazine administration. The final time-point of 120 min was chosen for convenience, to avoid the restriction of the horses in the stall for longer periods. The sum of the scores of the degree of sedation according to the tactile, auditory, and visual stimuli recorded at the experimental moment only for Phase II, was not publish elsewhere. Each stimulus was scored from 0 to 3, where zero represented no sedation and 3 the deepest sedation.

Four evaluators, experienced with sedation in horses, who did not communicate with each other observed and scored the two photos (lateral and oblique cranio-caudal) of the face of the horse at each moment. These were the responsible researcher (A.R.O.–Evaluator 1, E1) and three from different institutions holding a Diploma by the European College of Veterinary Anaesthesia and Analgesia [M.G.M.–Evaluator 2, E2 (since 2014); S.K.R.–Evaluator 3, E3 (since 2009); and S.S.–Evaluator 4, E4 (since 2009)]. All the evaluators were aware of the goal of the coding. The FaceSed was not analysed or scored on-site. Evaluators 1 and 2 were present on-site during the experiment and evaluator 1 was responsible for selecting the photos. Although this evaluator cannot be considered completely blinded, she was not aware of the treatment and moments, and it would be almost impossible in practical terms to memorise the 168 selected photos.

Prior to the start of the analysis, the evaluators were trained by evaluating 16 pairs of photos randomly chosen (10% of the total number of photos evaluated = 168). The pairs of photos were selected from four horses at each moment of each detomidine treatment by the main author—evaluator 1. The pairs of photos were randomised not only for the moments of each animal in sequence, but for the animals, time-points, and treatments. The photos were evaluated twice independently by each evaluator with an interval of one month. The randomisations were performed twice for the two observations within a one-month interval. After confirming high ($\geq$ 80%) intra (comparison between the first and second evaluation for each observer) and interobserver (matrix Spearman correlation comparison among observers at the second evaluation) correlations [46] regarding the attribution of scores, the main evaluations were started. This analysis was performed to guarantee that the evaluators would be reliable and genuinely involved in the study.

The 168 moments (pair of photos—lateral and oblique cranio-caudal) were randomised in the same way as in training. In this manner the evaluators scored the photos in two stages, each with different randomisation, at least one month apart. Evaluation spreadsheets and guidelines (S2 Appendix) for completing the data were made available. Evaluators were asked not to work on the project for more than one hour a day to avoid fatigue and to score first the numerical rating scale from 0 (not sedated) to 10 (maximum sedation) [39], followed by the FaceSed (S2 Appendix).

## Development of the FaceSed scale

The initial development of the FaceSed was adapted and modified from three out of six facial action units in horses undergoing pain compared to pain-free horses described in the horse grimace scale [31]. Only orbital aperture was coincident in the Grimace and FaceSed. The grimace scale description of the eye tightening corresponds to the eyelid partially or completely closed in the FaceSed. Strained mouth and lower lip with a pronounced chin from the grimace scale were modified to relaxation of the upper and lower lip. Stiff backwards ears from HGS were adapted to opening between the ear tips. In the present study, the grimace scale descriptions were adapted for the FaceSed to expressions indicative of muscle relaxation, characterised by the inability to sustain and move the facial action units. Other facial units from the grimace scale were excluded. After this first step comparing the grimace scale and FaceSed, other studies were assessed to find other possibly useful facial units that would call attention and might be easily identified by a human [26, 32, 33, 35, 36, 54]. The movements in the facial units of eyes, ears, and lips were identified according to EquiFACS [54]. However because none of the authors was EquiFACS certified, it was only used to identify possible indicators of movement of the facial musculature, which would resemble easily identifiable facial units described in other studies of horses under pain [32, 33] and sedation [26, 35, 36]. Other studies developing facial pain scales in horses suffering colic or noxious stimuli [32, 33] were also considered for the evaluation of the eyes, ears, and lips. Evaluations for the eyes [26, 36], ears [26], and lips [35, 36] have also been explored in studies that investigated the effect of different sedation protocols. Based on these observations, the FaceSed was developed with three scores for each item. Descriptors were based on the expected expression of muscle relaxation of each facial action unit for absent, partially sedated, or obviously sedated.

The content validation was performed to identify if the descriptions of the items were clear and relevant according to the theme of the scale. Content validation was achieved in three steps: i) the use of facial characteristics already described in studies with sedation, stress, and pain in horses [31–33, 54, 55], ii) the analysis of the semantic clarity of the content of the scale by the four principal evaluators before starting the training and main evaluation, and iii) the analysis of the relevance, performed by the three external experienced veterinary anaesthesiologists (M.O.T., F.A.O. and C.L.), not involved in the study, who attributed to the importance of each item on the scale as relevant +1, did not know how to give an opinion 0, or irrelevant -1. The relevance value attributed by each veterinary anaesthesiologist was summed and divided by three. The range of each score is -1 to +1, and values greater than 0.5 were considered relevant [56].

After the content was validated, the 168 photos were analysed twice by each of the four main evaluators and the analysis described below was performed. Only HHAG% data were collected on-site and published before [11, 48].

## Statistical analyses

**Reliability.** The reliability of the numerical rating scale and individual FaceSed items was calculated using the weighted kappa coefficient ($K_w$) and the FaceSed sum by the intraclass correlation coefficient (ICC) of the agreement type (Confidence interval). For repeatability, data from the first evaluation was compared with the second evaluation for each evaluator. For reproducibility, the scores of the first and second evaluations of all evaluators were compared using an agreement matrix. Weighted kappa and ICC were interpreted as very good (0.81–1.0), good (0.61–0.80), moderate (0.41–0.60), reasonable (0.21) 0.40), or poor (< 0.2) [46, 57].

The following analyses were performed with data from all evaluators, treatments, and grouped moments. The exceptions are described below.

**Concurrent criteria validation.** The most common way to test concurrent criteria validation is to correlate the proposal instrument with a gold-standard one aiming to measure the same purpose [46]. However, because when the study was performed there was no validated or gold standard scales to evaluate sedation in horses, the numerical rating scale was used, as it is a simple and intuitive scale and the HHAG% was used because is an objective measurement often used for several studies assessing sedation in horses. For this analysis treatments were grouped according to their similarities in sedation intensity, to evaluate any differences between: i) tranquilisation (ACPL + ACPH) and ii) sedation with low (DL + DLM) and iii) or high detomidine doses (DH + DHM). To test the validity of the concurrent criterion, the Spearman correlation was measured between the FaceSed with numerical rating scale and HHAG%, with the interpretation 0–35—low correlation; 0.35–0.7—mean correlation; 0.7–1.0 —high correlation [46, 56, 58].

**Construct validity (responsiveness).** The construct validity investigates whether the scale is measuring what it is set out to measure. Therefore, for responsiveness, it is expected that when horses are deeply sedated, their sedation scores should be higher than when sedation is abating or absent [56]. The data did not pass the Shapiro-Wilk normality test for responsiveness. Therefore, a Friedman test was used for all variables to evaluate differences over time (baseline, peak sedation, intermediate, and end of sedation) for each treatment and the Kruskal-Wallis test to compare the treatments within each moment.

**Principal component analysis.** The principal component analysis was used to define the dimensions of the scale according to the distribution of the items and how they correlate to each other [58]. Its interpretation was based on the Kaiser criterion [59], where eigenvalues > 1 and variance > 20 are approved and items on the scale with load factor $\geq 0.50$ or $\leq -0.50$ are selected. The eigenvalues and variance are coefficients extracted from the correlation matrix of the principal component analysis that indicate the degree of contribution of each dimension, to select only the representative dimensions [60].

**Internal consistency.** Another method used to evaluate the intercorrelation among items of the FaceSed was the internal consistency by Cronbach's α coefficient [61]. Minimally acceptable values are between 0.60–0.64, acceptable 0.65–0.69, good 0.70–0.74, very good 0.75–0.80, and excellent above 0.80 [62].

**Item-total correlation.** To find out if the items contributed to the total score of the scale in a homogeneous manner, Spearman's item-total correlation was performed between each item and the total sum of FaceSed after excluding the evaluated item. The value of each item in this analysis is interpreted as the individual relevance of the item compared to the total score of the scale. Acceptable values are between 0.30 and 0.70 [46].

**Sensitivity and specificity.** This analysis is mainly performed to help identifying the diagnostic accuracy. To calculate the sensitivity and specificity of a new test, it should be compared to a gold standard test to identify the true positives and negatives among what is being measured. Because there is no validated scale to evaluate sedation in horses, the sensitivity and specificity were calculated considering the presence (peak of sedation) or absence (at the baseline) of sedation. In the case of sedation, sensitivity ascertains if the instrument identifies the true positives (high sedation scores compatible with deep sedation) and specificity determines the true negatives (absent sedation scores compatible with non-sedated animals). Sensitivity towards detecting the presence of sedation (regardless of the degree) was calculated by the ratio between horses with scores $\geq 1$ at the peak of sedation (considered sedated or true positive) and the total number of horses, and, in a similar way, specificity by the relationship between horses with a score of 0 at baseline (not sedated or true negative) and the total number of horses. The interpretations are considered excellent when 95–100%, good when 85–94.9%, moderate when 70–84.9%, and non-sensitive or specific when <70% [39, 63].

**ROC curve for determination of the cut-off point of the FaceSed.** The Receiver Operating Characteristic (ROC) curve is the graphical representation of the relationship between sensitivity at the peak of sedation (to detect the truly sedated horses) and specificity at the baseline time-point (to distinguish the truly negative non-sedated horses). The discriminatory capacity of the test is determined by the area under the curve [64]. Only horses treated with high doses of detomidine were used for this analysis. The HHAG% was used in this analysis as a predictive value because it is the most commonly used objective sedation measure in horses. Horses with an HHAG% ≤50% were considered truly sedated, because this is the value used in some studies to consider horses sufficiently sedated for standing procedures [4, 9]. To determine the cut-off point of the scale, the Younden index and the diagnostic uncertainty zone were defined. The Younden index is the coincident score with the highest sensitivity and specificity of the scale according to the ROC curve, and the diagnostic uncertainty zone or grey zone is the diagnostic accuracy when calculating 95% CI by replicating the original ROC curve 1001 times using the bootstrap method and the sensitivity and specificity value > 90%. This interval of the lowest and the highest values of these two methods is the grey zone [46, 65].

**Frequency distribution of scores.** Finally, the frequency of sedation scores that were assigned by the evaluators was performed in the four evaluation time-points, within each grouped treatment of tranquilisation (ACPL + ACPH) and sedation with low (DL + DLM) and high doses of detomidine (DH + DHM). This analysis was performed to assess the presence or absence of each score attributed to each item in horses under different depths of sedation or in the state of normality, to investigate their representativeness and importance.

The statistical analysis in this manuscript was performed using R software in the RStudio integrated development environment (RStudio Team– 2016) and Microsoft Office® (Excel—2019). The statistical significance was accepted at p<0.05.

## Results

The FaceSed showed intra and interobserver reliability, content, criterion and construct validities, homogeneity (item-total correlation), good internal consistency and sensitivity to assess sedation in horses.

### Content validity

The semantic clarity of the content of the scale of the characteristics described at FaceSed based on previous studies [31, 33, 44, 54] was approved (Table 1 and Fig 1).

The item partial opening between the tips of the ears or asymmetry representative of score 1 presented a mean of less than 0.5 for the degree of relevance (Table 1). All sub-items of the upper lip relaxation had a mean lower than 0.5 by the external assessors, but it was maintained because these characteristics have been described and used in previous studies.

### Intraobserver reliability (repeatability)

The repeatability (ICC) of the sum of the FaceSed of all the four observers together ranged from good to very good (0.74–0.94) (Table 2). The repeatability of the numerical rating scale was very good (0.86–0.92).

### Interobserver reliability (reproducibility)

The numerical rating scale showed very good agreements (0.83–0.88), while the FaceSed showed moderate to very good agreements (0.57–0.87).

**Table 1. Content validity of FaceSed developed to evaluate the degree of sedation in horses.**

| Area evaluated | Relaxation Intensity | Scores | Relevance (-1, 0, 1) | References |
|---|---|---|---|---|
| **Ears** | | | | [31, 33, 44, 54] |
| | No opening between the ear tips, position of attention | 0 | 0.66 | |
| | Partial opening between the ear tips or asymmetry | 1 | 0.33 | |
| | Wide opening between the ear tips (ears relaxed) | 2 | 0.66 | |
| **Eyes (orbital opening)** | | | | [31, 33, 44, 54] |
| | Eyes completely opened | 0 | 1 | |
| | Eyes partially opened | 1 | 1 | |
| | Eyes almost or completely closed | 2 | 1 | |
| **Relaxation of the lower lip** | | | | [33, 44, 54] |
| | No signs of lower lip relaxation and/or closed mouth | 0 | 1 | |
| | Slight relaxation of lower lip | 1 | 1 | |
| | Pronounced relaxation of lower lip and/or open mouth | 2 | 1 | |
| **Relaxation of the upper lip** | | | | |
| | No signs of upper lip relaxation | 0 | 0.33 | |
| | Slight upper lip relaxation | 1 | 0 | |
| | Pronounced upper lip relaxation | 2 | 0.33 | |

FaceSed—Numerical Facial Scale of Sedation in Horses. Values greater than 0.5 are considered relevant.

The item orbital opening presented good to very good agreement (0.68–0.88) and the others presented reasonable to good agreement (0.26–0.71) between observers (Table 3).

## Concurrent criteria validation

The correlations of the sum of FaceSed with the numerical rating scale for the animals treated with acepromazine, low and high detomidine doses and all groups together were 0.85, 0.91, 0.95, and 0.92, respectively and for HHAG% were -0.56, -0.77, -0.80, and -0.75, respectively.

## Construct validity (responsiveness)

All FaceSed items, their sum, the numerical rating scale and HHAG% (except ACPL) presented higher scores at peak sedation/tranquilisation for all treatments than baseline (Table 4).

When analysing all grouped treatments, the FaceSed and numerical rating scale scores at the end of sedation were significantly different from baseline (Fig 2). However, when evaluating the treatments alone, it was observed that the scores for the sum of Facesed at the end of sedation were only higher compared to baseline in groups ACPL, ACPH and DHM, while for the numerical rating scale, this was only true in groups ACPL and ACPH (Table 4).

When comparing treatments at peak sedation, the FaceSed and numerical rating scale in DL and DLM groups were significantly different from DH and DHM respectively, and the treatments DH and DHM were significantly different from the acepromazine treatments (Table 4).

## Principal component analysis

The multiple association analysis by principal components (Table 5) defined the scale as unidimensional, as it presented the largest load factors for each item in the first dimension (> 0.5), with eigenvalue > 1 and variance > 20 [59].

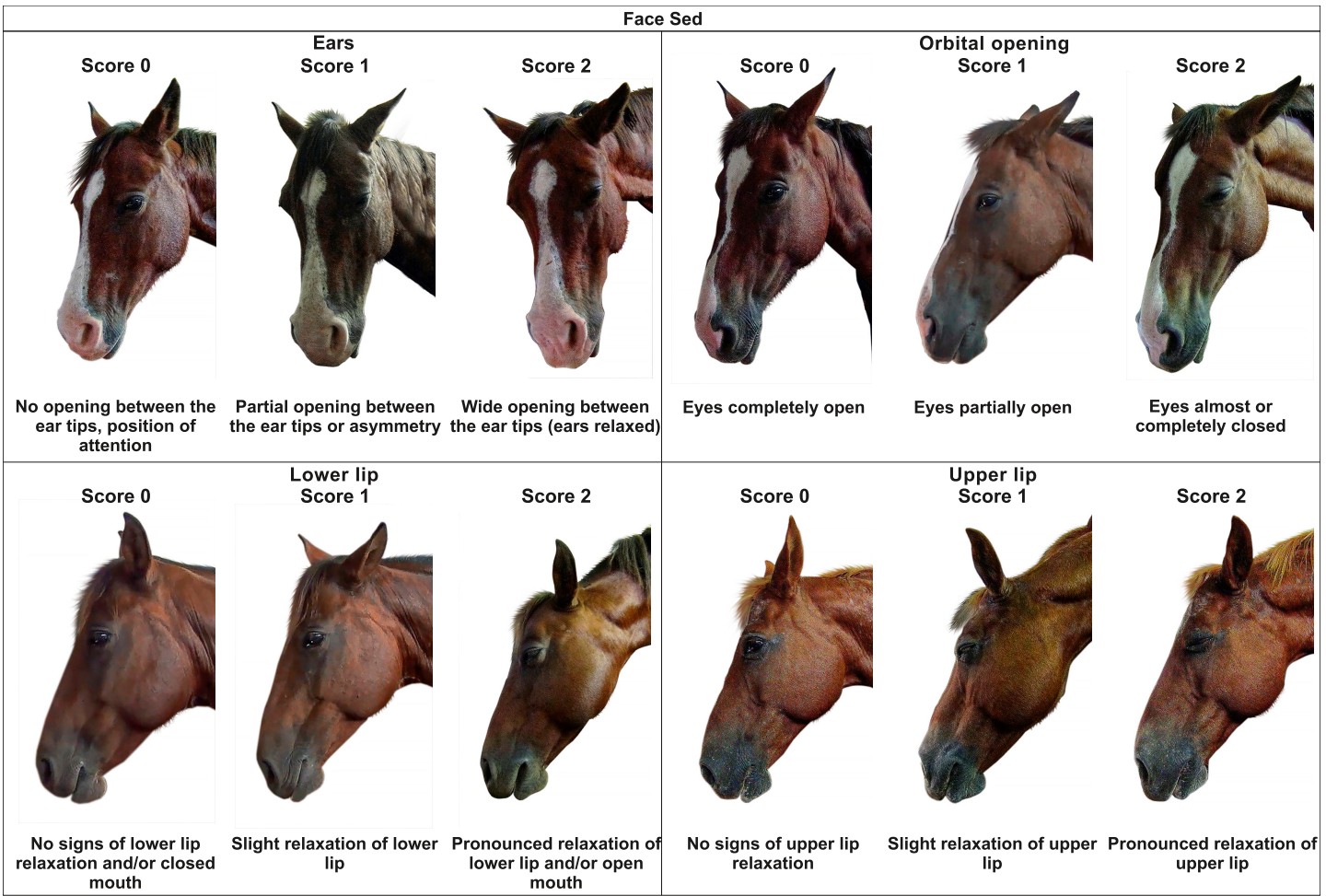

**Fig 1. Facial sedation scale in horses (FaceSed).**

**Table 2. Intra-rater reliability of FaceSed and numerical rating scales between the first and second observations (confidence interval).**

| FaceSed Items | E1 | | E2 | | E3 | | E4 | |
|---|---|---|---|---|---|---|---|---|
| | $k_w$ | CI | $k_w$ | CI | $k_w$ | CI | $k_w$ | CI |
| Ears | 0.8 | 0.73–0.86 | 0.71 | 0.71–0.71 | 0.56 | 0.48–0.64 | 0.71 | 0.71–0.71 |
| Orbital opening | 0.85 | 0.79–0.91 | 0.85 | 0.8–0.9 | 0.72 | 0.67–0.77 | 0.86 | 0.86–0.86 |
| Lower lip | 0.72 | 0.68–0.76 | 0.62 | 0.53–0.7 | 0.68 | 0.62–0.75 | 0.7 | 0.66–0.74 |
| Upper lip | 0.77 | 0.77–0.77 | 0.7 | 0.67–0.73 | 0.62 | 0.53–0.71 | 0.77 | 0.71–0.82 |
| NRS | 0.9 | 0.9–0.9 | 0.86 | 0.86–0.86 | 0.91 | 0.91–0.91 | 0.92 | 0.92–0.92 |
| | ICC | CI | ICC | CI | ICC | CI | ICC | CI |
| FaceSed | 0.91 | 0.89–0.94 | 0.86 | 0.74–0.90 | 0.82 | 0.78–0.86 | 0.89 | 0.85–0.92 |

FaceSed—Numerical facial scale of sedation in horses. NRS -Numerical rating scale. E1—Evaluator 1, E2—Evaluator 2, E3—Evaluator 3, E4—Evaluator 4. Interpretation of weighted Kappa (*Kw*) and Intraclass correlation coefficient (ICC)—very good 0.81–1.0; good 0.61–0.80; moderate 0.41–0.60, reasonable 0.21–0.40; and poor < 0.20. Confidence interval (CI) [46, 57, 58].

**Table 3. Inter-rater matrix comparison of FaceSed and numerical rating scale scores among all evaluators.**

| Evaluator | E1 | | E2 | | E3 | |
|---|---|---|---|---|---|---|
| | $k_w$ | CI | $k_w$ | CI | $k_w$ | CI |
| **Ears** | | | | | | |
| E2 | **0.39** | 0.33–0.46 | | | | |
| E3 | **0.55** | 0.47–0.62 | **0.54** | 0.46–0.62 | | |
| E4 | **0.61** | 0.53–0.69 | **0.57** | 0.48–0.65 | **0.61** | 0.53–0.68 |
| **Orbital opening** | | | | | | |
| E2 | **0.83** | 0.79–0.88 | | | | |
| E3 | **0.74** | 0.68–0.80 | **0.75** | 0.71–0.79 | | |
| E4 | **0.80** | 0.75–0.84 | **0.84** | 0.82–0.86 | **0.75** | 0.71–0.78 |
| **Lower lip** | | | | | | |
| E2 | **0.44** | 0.26–0.63 | | | | |
| E3 | **0.50** | 0.42–0.57 | **0.45** | 0.37–0.53 | | |
| E4 | **0.64** | 0.59–0.70 | **0.49** | 0.40–0.59 | **0.45** | 0.37–0.52 |
| **Upper lip** | | | | | | |
| E2 | **0.65** | 0.59–0.71 | | | | |
| E3 | **0.54** | 0.47–0.61 | **0.52** | 0.45–0.59 | | |
| E4 | **0.65** | 0.58–0.71 | **0.64** | 0.59–0.70 | **0.59** | 0.52–0.65 |
| **NRS** | | | | | | |
| E2 | **0.83** | 0.83–0.83 | | | | |
| E3 | **0.85** | 0.85–0.85 | **0.83** | 0.83–0.83 | | |
| E4 | **0.88** | 0.88–0.88 | **0.83** | 0.83–0.83 | **0.83** | 0.83–0.83 |
| | ICC | CI | ICC | CI | ICC | CI |
| **FaceSed** | | | | | | |
| E2 | **0.75** | 0.70–0.79 | | | | |
| E3 | **0.64** | 0.57–0.70 | **0.73** | 0.67–0.77 | | |
| E4 | **0.84** | 0.80–0.87 | **0.82** | 0.78–0.85 | **0.71** | 0.65–0.76 |

FaceSed—Numerical facial scale of sedation in horses. NRS—numerical rating scale. E1—Evaluator 1, E2—Evaluator 2, E3—Evaluator 3, E4—Evaluator 4. Interpretation of weighted Kappa (Kw) and Intraclass correlation coefficient (ICC)—very good 0.81–1.0; good 0.61–0.80; moderate 0.41–0.60, reasonable 0.21–0.40; and poor < 0.20. Confidence interval (CI) [46, 57, 58].

The items of the scale presented all vectors direction identifying intermediate and deep sedation (Fig 3).

### Item-total correlation and internal consistency

The item-total correlation fell within the values considered acceptable, from 0.3 to 0.7 [46], except for the item eyes (0.73), and the internal consistency was excellent for all items on the scale (Table 6).

### Sensitivity and specificity

All items presented sensitivity to the isolated or grouped treatments, however, specificity was only observed for the items orbital opening and upper lip relaxation (Table 7).

### ROC curve and cut-off point of the FaceSed

The area under the curve was 0.96, representing the high precision of the scale (Fig 4). The Younden index was > 5 for all the evaluators. The resampling bootstrap CIs were 4.5 and 5.5,

**Table 4. Responsiveness of FaceSed, numerical rating scale, and HHAG% over time and between treatments.**

| FaceSed | | Moments | | | | | | | |
|---|---|---|---|---|---|---|---|---|---|
| | | **Baseline** | | **Peak of sedation** | | **Intermediate** | | **End of sedation** | |
| Items | | Median | Amplitude | Median | Amplitude | Median | Amplitude | Median | Amplitude |
| Ears | ACPL | $0.5^{bB}$ | 0–2 | $1^{aB}$ | 0–2 | $1^{aB}$ | 0–2 | $1^{abAB}$ | 0–2 |
| | ACPH | $1^{bA}$ | 0–2 | $1^{aB}$ | 0–2 | $1^{aB}$ | 0–2 | $1^{aA}$ | 0–2 |
| | DL | $1^{cAB}$ | 0–2 | $2^{aA}$ | 0–2 | $1^{bB}$ | 0–2 | $1^{bcAB}$ | 0–2 |
| | DLM | $0^{cB}$ | 0–2 | $2^{aA}$ | 0–2 | $1^{bB}$ | 0–2 | $1^{cB}$ | 0–2 |
| | DH | $1^{bAB}$ | 0–2 | $2^{aA}$ | 0–2 | $2^{aA}$ | 0–2 | $1^{bAB}$ | 0–2 |
| | DHM | $0^{bB}$ | 0–2 | $2^{aA}$ | 0–2 | $2^{aA}$ | 0–2 | $1^{bB}$ | 0–2 |
| | All | $1^{d}$ | 0–2 | $2^{a}$ | 0–2 | $2^{b}$ | 0–2 | $1^{c}$ | 0–2 |
| Orbital opening | ACPL | $0^{b}$ | 0–1 | $1^{aD}$ | 0–2 | $1^{aC}$ | 0–2 | $1^{aB}$ | 0–2 |
| | ACPH | $0^{b}$ | 0–1 | $1^{aCD}$ | 0–2 | $1^{aBD}$ | 1–2 | $1^{aA}$ | 0–2 |
| | DL | $0^{c}$ | 0–1 | $2^{aAB}$ | 1–2 | $1^{bCD}$ | 0–2 | $0^{cC}$ | 0–1 |
| | DLM | $0^{c}$ | 0–1 | $1^{aBC}$ | 0–2 | $1^{bC}$ | 0–2 | $0^{cC}$ | 0–1 |
| | DH | $0^{b}$ | 0–2 | $2^{aA}$ | 1–2 | $1.5^{aAB}$ | 1–2 | $0^{bC}$ | 0–1 |
| | DHM | $0^{b}$ | 0–1 | $2^{aA}$ | 0–2 | $2^{aA}$ | 0–2 | $0^{bBC}$ | 0–2 |
| | All | $0^{d}$ | 0–2 | $2^{a}$ | 0–2 | $1^{b}$ | 0–2 | $1^{c}$ | 0–2 |
| Lower Lip | ACPL | $0^{b}$ | 0–1 | $1^{aB}$ | 0–2 | $1^{aAB}$ | 0–2 | $1^{aAB}$ | 0–2 |
| | ACPH | $0.5^{b}$ | 0–2 | $1^{aB}$ | 0–2 | $1^{aA}$ | 0–2 | $1^{aA}$ | 0–2 |
| | DL | $0^{c}$ | 0–2 | $1^{aAB}$ | 0–2 | $1^{bcB}$ | 0–2 | $1b^{cBC}$ | 0–2 |
| | DLM | $1^{b}$ | 0–2 | $1^{aB}$ | 0–2 | $1^{abB}$ | 0–2 | $0^{bC}$ | 0–2 |
| | DH | $1^{c}$ | 0–2 | $2^{aA}$ | 1–2 | $1^{bA}$ | 0–2 | $0^{cC}$ | 0–2 |
| | DHM | $1^{b}$ | 0–2 | $2^{aA}$ | 0–2 | $1^{aA}$ | 0–2 | $0^{bC}$ | 0–1 |
| | All | $1^{c}$ | 0–2 | $1^{a}$ | 0–2 | $1^{b}$ | 0–2 | $1^{c}$ | 0–2 |
| Upper Lip | ACPL | $0^{b}$ | 0–1 | $1^{aC}$ | 0–2 | $1^{aB}$ | 0–2 | $0.5^{a}$ | 0–2 |
| | ACPH | $0^{c}$ | 0–1 | $1^{abBC}$ | 0–2 | $1^{aA}$ | 0–2 | $1^{bc}$ | 0–2 |
| | DL | $0^{c}$ | 0–2 | $1^{aB}$ | 0–2 | $0^{bcB}$ | 0–2 | $0^{bc}$ | 0–2 |
| | DLM | $0^{c}$ | 0–1 | $1^{aBC}$ | 0–2 | $1^{abB}$ | 0–2 | $0^{bc}$ | 0–2 |
| | DH | $0^{c}$ | 0–2 | $2^{aA}$ | 0–2 | $1^{bA}$ | 0–2 | $0^{c}$ | 0–2 |
| | DHM | $0^{b}$ | 0–1 | $2^{aA}$ | 0–2 | $1^{aA}$ | 0–2 | $0^{b}$ | 0–2 |
| | All | $0^{d}$ | 0–2 | $1^{a}$ | 0–2 | $1^{b}$ | 0–2 | $0^{c}$ | 0–2 |
| FaceSed | ACPL | $1^{c}$ | 0–5 | $4^{aD}$ | 1–8 | $3^{abC}$ | 1–7 | $3^{bAB}$ | 0–7 |
| | ACPH | $2^{c}$ | 0–5 | $4^{abD}$ | 2–8 | $5^{aB}$ | 2–8 | $4^{bA}$ | 1–8 |
| | DL | $2^{c}$ | 0–5 | $6^{aBC}$ | 3–8 | $3^{bC}$ | 0–8 | $2^{bcBC}$ | 0–5 |
| | DLM | $1^{c}$ | 0–6 | $5^{aCD}$ | 1–8 | $4^{bC}$ | 0–8 | $2^{cC}$ | 0–6 |
| | DH | $2^{c}$ | 0–8 | $7^{aA}$ | 4–8 | $6^{bAB}$ | 1–8 | $2^{cC}$ | 0–7 |
| | DHM | $1^{b}$ | 0–4 | $7^{aAB}$ | 4–8 | $6^{aA}$ | 1–8 | $2^{bBC}$ | 0–5 |
| | All | $1,5^{d}$ | 0–8 | $6^{a}$ | 1–8 | $4^{b}$ | 0–8 | $2^{c}$ | 0–8 |
| NRS | ACPL | $2^{bB}$ | 0–5 | $4^{aE}$ | 2–10 | $4^{aC}$ | 2–7 | $4^{aB}$ | 0–7 |
| | ACPH | $2^{bA}$ | 1–5 | $6^{aDE}$ | 3–8 | $5.5^{aB}$ | 3–10 | $5^{aA}$ | 2–9 |
| | DL | $2^{cAB}$ | 0–4 | $7^{aBC}$ | 4–10 | $4^{bC}$ | 1–10 | $3^{cBC}$ | 1–6 |
| | DLM | $2^{cAB}$ | 0–6 | $6^{aCD}$ | 3–10 | $4^{bC}$ | 1–10 | $2^{cC}$ | 0–5 |
| | DH | $2^{cAB}$ | 0–10 | $9^{aA}$ | 5–10 | $6.5^{bAB}$ | 2–10 | $3^{cC}$ | 1–8 |
| | DHM | $2^{bB}$ | 0–5 | $9^{aAB}$ | 4–10 | $7^{aA}$ | 2–10 | $3^{bC}$ | 1–6 |
| | All | $2^{d}$ | 0–10 | $7^{a}$ | 2–10 | $5^{b}$ | 1–10 | $3^{c}$ | 0–9 |

*(Continued)*

**Table 4.** (Continued)

| FaceSed | | Moments | | | | | | | |
|---|---|---|---|---|---|---|---|---|---|
| | | Baseline | | Peak of sedation | | Intermediate | | End of sedation | |
| Items | | Median | Amplitude | Median | Amplitude | Median | Amplitude | Median | Amplitude |
| HHAG%* | ACPL | 100[a] | 100–100 | 89[abA] | 27–100 | 84[bAB] | 81–90 | 95[ab] | 71–103 |
| | ACPH | 100[a] | 100–100 | 66[bAB] | 53–91 | 82[abAB] | 50–102 | 92[ab] | 68–105 |
| | DL | 100[a] | 100–100 | 53[bAB] | 22–71 | 93[abA] | 44–105 | 103[a] | 83–110 |
| | DLM | 100[a] | 100–100 | 64[bAB] | 29–89 | 85[abAB] | 47–95 | 100[a] | 96–107 |
| | DH | 100[a] | 100–100 | 27[bB] | 18–47 | 60[abB] | 27–80 | 100[a] | 86–105 |
| | DHM | 100[a] | 100–100 | 29[bB] | 14–74 | 68[abAB] | 26–95 | 102[a] | 84–147 |
| | All | 100[a] | 100–100 | 53[b] | 14–100 | 82[b] | 26–105 | 100[a] | 68–147 |

FaceSed—Numerical facial scale of sedation in horses, NRS—Numerical rating scale, *HHAG%—Head height above the ground (data collected in a previously published study in cm [11, 48] doi:10.1111/evj.13054 and doi:10.3389/fvets.2021.611729 respectively), ACPL + ACPH (acepromazine in low and high doses); DL + DLM (low dose detomidine and associated with methadone); DH + DHM (high dose detomidine and associated with methadone). Different lower-case letters represent statistical differences over time (p <0.05) (a> b> c). Different capital letters represent statistical differences between treatments (p <0.05) (A>B>C).

and the ranges of sensitivity and specificity > 90 were 5.3 and 5.5. Based on the resampling result, the diagnostic uncertainty zone ranged from 5.3 to 5.5, which means that horses with scores < 5 are not sufficiently sedated and horses with scores > 6 are sufficiently sedated.

## Frequency distribution of scores

There was a higher frequency of 0 scores at baseline for the eyes and upper lip items (Fig 5), corroborating with the specificity data (Table 7).

Scores 1 and 2 predominated at peak sedation and/or intermediate sedation for all items.

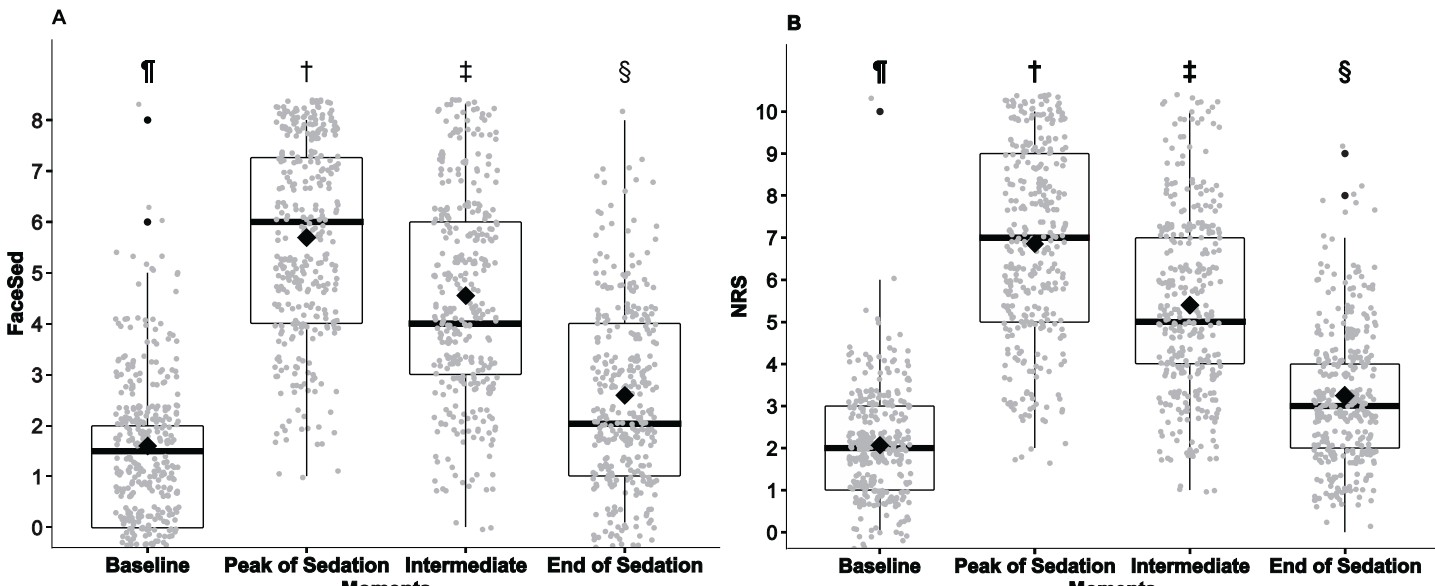

**Fig 2. FaceSed (A) and numerical rating scale (B) scores before and after grouped treatments (ACPL, ACPH, DL, DLM, DH, and DHM).** ACPL + ACPH—low and high dose acepromazine, DL + DLM—low dose detomidine and associated with methadone, DH and DHM—high dose detomidine and associated with methadone. Different symbols indicate significant differences between them (p<0.05). † > ‡> § >¶.

**Table 5. Load values, eigenvalues, and variance of FaceSed items by principal component analysis.**

| FaceSed Items | Load factors in Dimension 1 | Load factors in Dimension 2 |
|---|---|---|
| Ears | **0.78** | 0.52 |
| Orbital opening | **0.86** | 0.11 |
| Lower lip relaxation | **0.77** | -0.51 |
| Upper lip relaxation | **0.83** | -0.13 |
| **Eigenvalue** | **2.63** | 0.56 |
| **Variance** | **65.72** | 14.12 |

FaceSed—Numerical facial scale of sedation in horses. Values in bold represent eigenvalues > 1, variance > 20, and load factor ≥ 0.50 or ≤ -0.50 approved according to the Kaiser criterion [59].

## Discussion

The results of this study show that FaceSed is a simple and practical scale that offers reliability and validity to evaluate sedation over time in horses submitted to tranquilisation with acepromazine and sedation with the alpha-2 agonist detomidine with or without the opioid methadone, based on the validation criteria of the literature [39, 40, 56, 66–68]. This main advantage is that it does not imply interaction with the horse. However, differentiation between

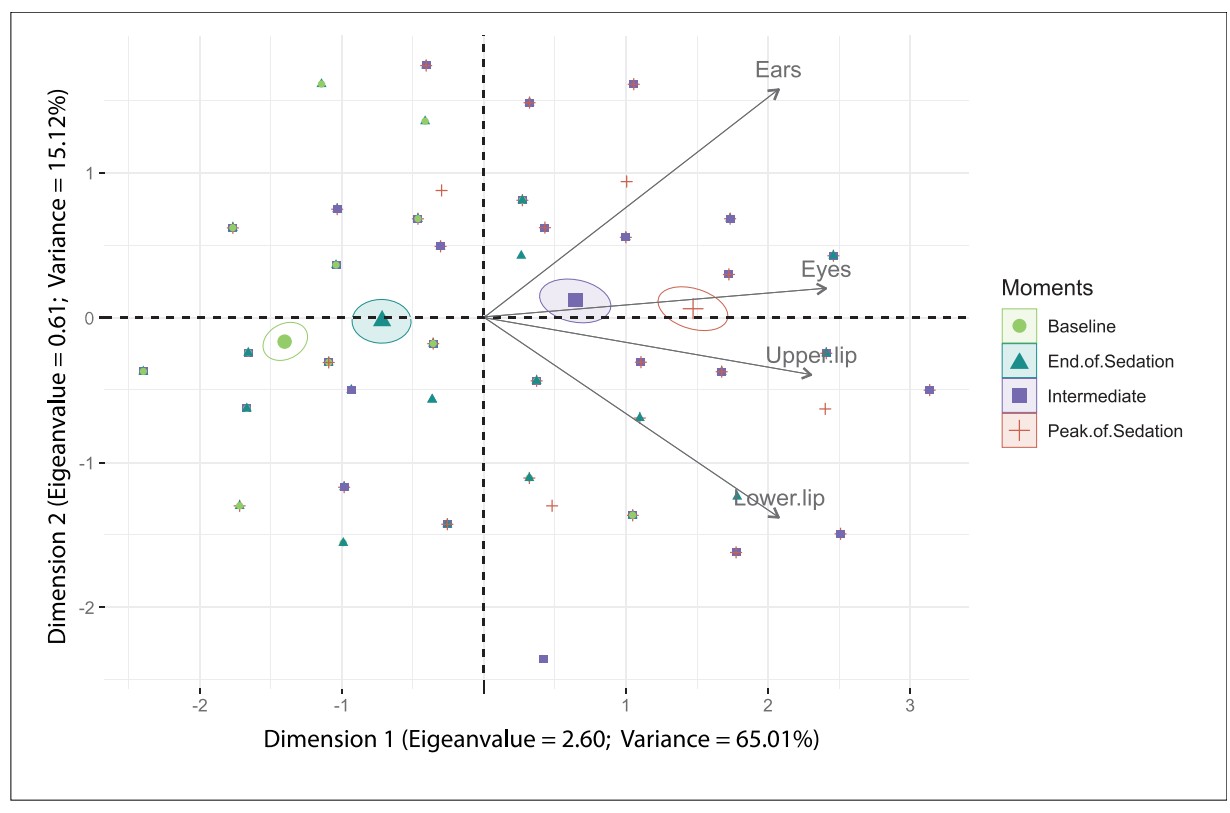

**Fig 3. FaceSed principal component analysis biplot.** Confidence ellipses were built according to the moments before and after sedation. Baseline (green); Peak of sedation (red); Intermediate (purple); End of sedation (blue). The ellipses on the left represent the absence or end of sedation and on the right represent the peak or intermediate sedation. The time-points peak and intermediate sedation influence all items on the scale since their vectors are directed to these ellipses.

**Table 6. Spearman item-total correlation and internal consistency of each FaceSed item.**

| FaceSed Item | Item-total Correlation | Internal Consistency |
|---|---|---|
| All items | | **0.83** |
| Excluding Ears | **0.60** | **0.80** |
| Excluding Orbital opening | **0.73** | **0.74** |
| Excluding Lower lip relaxation | **0.60** | **0.80** |
| Excluding Upper lip relaxation | **0.68** | **0.77** |

FaceSed—Numerical facial scale of sedation in horses. Interpretation of Spearman's item-total correlation (rs)— values between 0.3 and 0.7 are accepted and stand out in bold [46]. Cronbach's α coefficient was calculated by the total score of the scale and excluding each item from the scale. Interpretation: minimally acceptable 0.60–0.64, acceptable 0.65–0.69, good 0.70–0.74, very good 0.75–0.80, and excellent > 0,80 [62]. **Bold values are above 0.70**.

tranquilisation and low degree of sedation may be difficult to assess using both FaceSed and numerical rating scale.

The training phase that our evaluators faced resulted in a good inter-observer correlation. Even when the evaluators might be experienced and familiarised with the scale, this is not a guarantee of good reliability [69]. Therefore, training is strongly recommended even when using validating scales as it improves reproducibility [70, 71].

**Table 7. FaceSed sensitivity and specificity.**

| FaceSed Items | Ears | Orbital opening | Lower lip relaxation | Upper lip relaxation |
|---|---|---|---|---|
| All treatments | | | | |
| Sensitivity % | 97 | 96 | 87 | 82 |
| Specificity % | 44 | 79 | 49 | 82 |

FaceSed—Numerical facial scale of sedation in horses. Interpretation: excellent 95–100%, good 85–94.9%, moderate 70–84.9%, and non-sensitive or non-specific < 70% [39, 63].

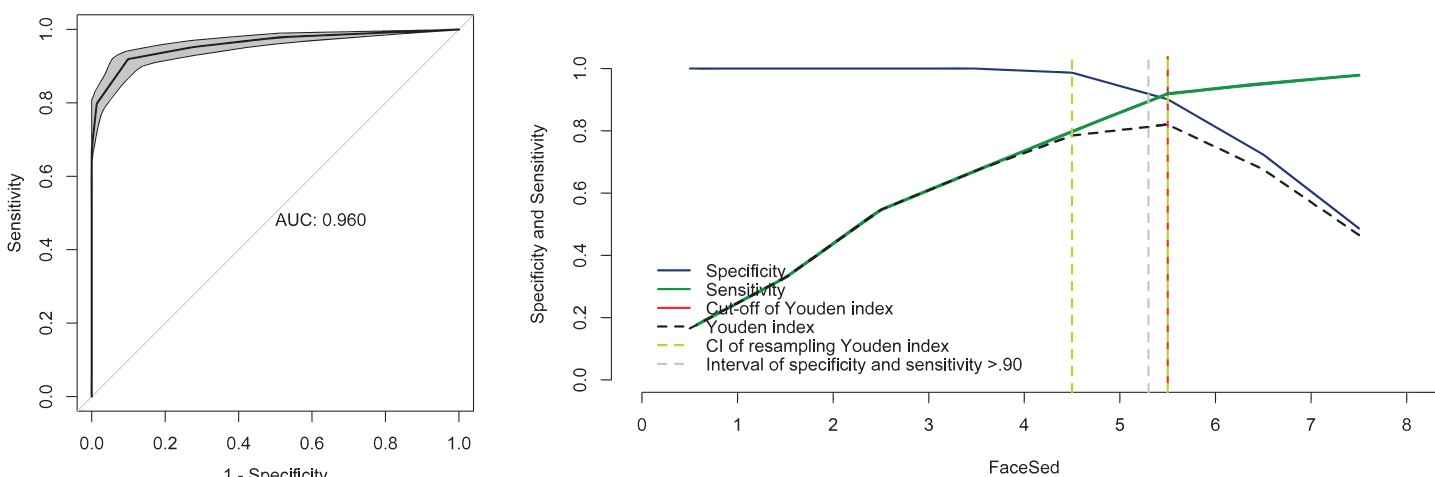

**Fig 4. Area under the curve (AUC) and two-graph ROC curve with the diagnostic uncertainty zone.** Interpretation of AUC: > 0.90 presents high discriminatory capacity. The two-graph ROC curve estimates the diagnostic uncertainty zone of the cut-off point according to the Youden index.

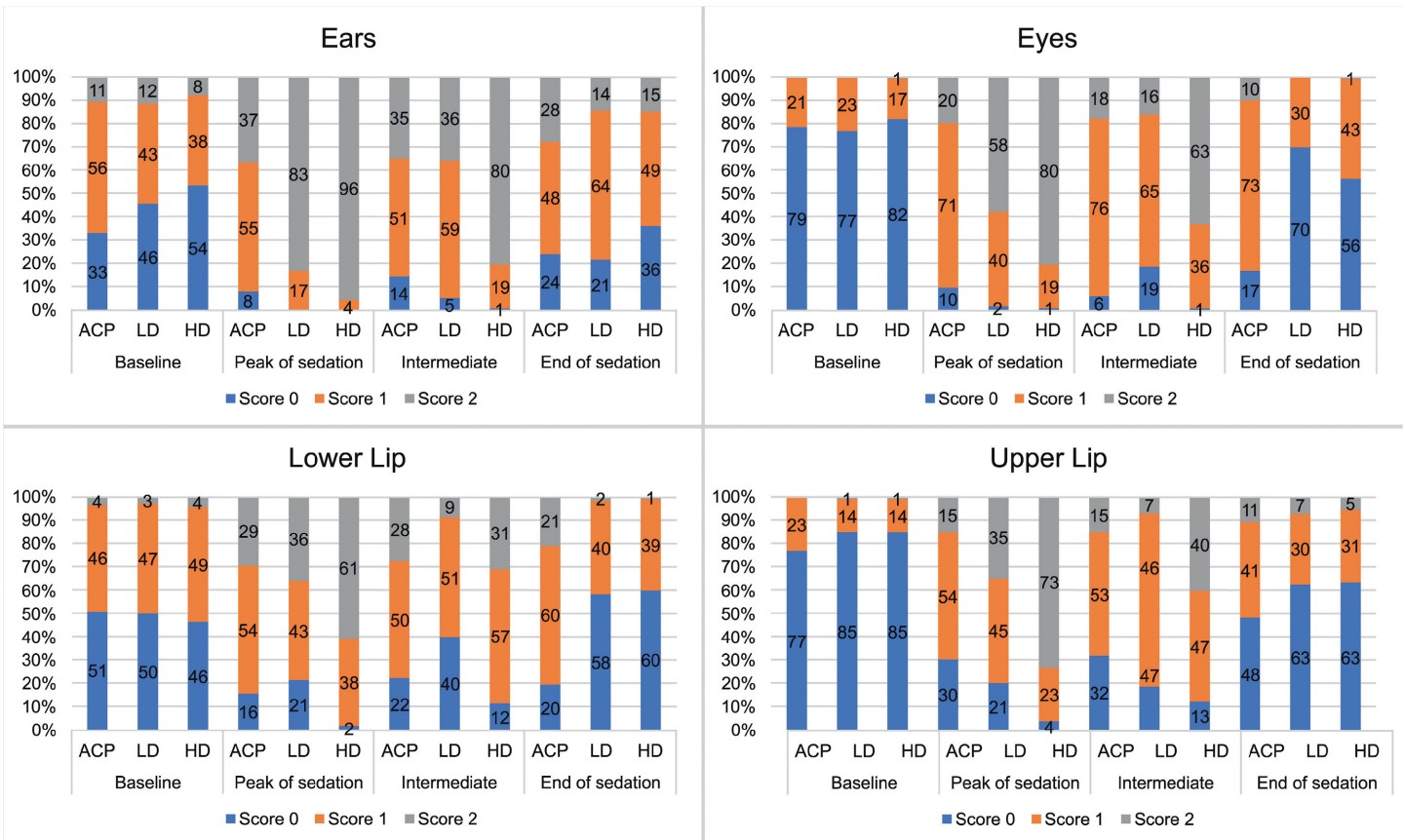

**Fig 5. Frequency distribution of FaceSed scores before and after sedation/tranquilisation.** ACPL + ACPH—acepromazine in low and high doses, DL + DLM—detomidine in low dose and associated with methadone, DH and DHM—detomidine in high dose and associated with methadone.

According to the results of content validation by the external veterinary anaesthesiologists, the relevance of score 1 for partial opening of the ears was less than 0.5. However, as the other ear-related items were approved, we decided to maintain it. The item upper lip relaxation was not considered relevant, possibly because the upper lip drop is not described in other studies evaluating sedation. Thus, the consultants' lack of familiarity may have led to the questioning of its relevance. Still, this item was maintained given to its importance during the development of the FaceSed and justified by the subsequent analysis. The content validity is used to identify how well item descriptions are formulated and cover the proposed theme [21] and in this study, FaceSed content validity was consolidated in a similar way of pain scales in animals [39, 41–43, 56].

Phase II was performed 11 months apart from phase I, because it was not parallel to the other previously published simultaneous studies [11, 49, 50]. We decided to also complement the validity assessment of the FaceSed for tranquilisation, and this was the only period that the facilities, horses, and authors were available. However all photos from phases I and II were combined and randomised for evaluation by the observers without distinction of phases.

The sum of the FaceSed presented good to very good intraobserver reliability and moderate to very good interobserver reliability, which guarantees the repeatability and reproducibility of the FaceSed for future studies. This is in contrast with a previous study that evaluated facial

expression in ridden horses [55]. That study suggested that reproducibility for eye evaluations was not very consistent, with Kappa values < 0.42 even with trained evaluators. This discrepancy between the two studies may be explained because the FaceSed is a simpler instrument with fewer and more easily identifiable descriptors of sedative expressions compared to facial expressions in ridden horses. Our results showed reduced interobserver reliability for the items ears, and lower and upper lips, which possibly contributed to the interobserver reliability of the FaceSed being slightly lower than that of numerical rating scale. The biases that could affect the reliability of an instrument are mainly a prolonged evaluation time, which makes the evaluators tired, inadequate description of the items, and finally, lack of practice of the evaluators [72, 73]. None of these biases were apparently the case in this study.

The fact that the numerical rating scale presented better inter-rater reliability than the FaceSed was surprising since the reliability of unidimensional scales is usually not good [25]. The numerical rating scale was scored before the FaceSed, which excluded the bias of scoring it based on the facial score. However, the fact that the evaluators were previously trained to recognise the facial sedation characteristics described by the FaceSed may have improved reliability of the numerical rating scale. Another point was that for some photos it was possible to identify sweating, and because neck position might be apparent in some of the lateral view pictures, the observers might be able to distinguish the low HHAG% induced by sedation. Thus, the FaceSed may have been influenced by the previous evaluation of the numerical rating scale and neck position for some photos. This bias could be eliminated if the evaluations were performed twice and independently for each scale, however this was not feasible considering the extensive data analysis. Otherwise, if the FaceSed was scored first, its sedation descriptors might facilitate the appraisal of the numerical rating scale, leading to possible overrating of its reliability. Another point is that the fact that the FaceSed and the numerical rating scale were sequentially scored may have overestimated their correlation, however this is the usual procedure for concurrent criterion validation of instruments in the literature [38–40, 56, 67, 68].

When proposing a new scale, the ideal method to perform concurrent criterion validation is to compare it with validated methods in order to find out how much their scores correlate with the proposed scale [74]. Since, unlike for equine pain scales [39, 44, 63], there is no validated instrument to evaluate sedation in horses, the numerical rating scale and the HHAG% were used. The approach based on correlation of the FaceSed against a unidimensional scale has been previously used when novel pain scales were elaborated for other species [41, 42, 56, 68]. Indeed, the FaceSed presented a high positive correlation with the numerical rating scale in all treatments, indicating the similarity in their magnitudes. The data on the HHAG% used in the present study have already been published in the simultaneous studies ran in parallel to the present one [11] and they were used here because the HHAG% is the most established and objective method to assess sedation in horses [9, 12, 14], and therefore the closest 'gold standard' instrument to compare with the FaceSed. There was a high negative correlation between the FaceSed and HHAG% for detomidine treatments and with all treatments combined and, as expected, an average correlation with acepromazine, as this drug does not reduce HHAG%.

Both the total FaceSed score and each of its items identified the facial changes of the horses over time both under tranquilisation and under sedation, since they presented higher scores at the peak and intermediate tranquilisation/sedation when compared to baseline and end of tranquilisation/sedation. The same was observed for numerical rating scale and HHAG%. These results are of importance as a responsive scale must detect the differences in scores in relation to the interventions that the instrument proposes to measure [75].

At the time-point of peak sedation, the FaceSed scores were lowest for tranquilisation (low and high dose of acepromazine: ACPL and ACPH), intermediate for moderate sedation with a low dose of detomidine (DL and DLM), and highest for deepest sedation with a high dose of

detomidine (DH and DHM). Otherwise HHAG%, a widely used measure to assess sedation, failed to differentiate tranquilisation with acepromazine from low and high sedation intensities with detomidine (with or without methadone), possibly because lowering of the head is not so evident in horses tranquilised with acepromazine.

In the principal component analysis all items of the FaceSed followed the same trend to measure sedation in horses, as they followed the Kaiser criterion and presented in only one dimension load factors ≥ 0.50, eigenvalue> 1, and variance> 20 [59]. Also, the directions of the vectors indicate that all items identified sedation. The practical meaning is that all items are influenced by the intensity of sedation. This multivariate analysis prevents the possibility of the inclusion of items that are not mutually associated [76].

Another way to evaluate the correlation between the items on the scale is the internal consistency by Cronbach's α coefficient, which was high for the FaceSed. This analysis can be interpreted together with the principal component analysis, in which high values of Cronbach's α coefficient usually occur in situations where few dimensions are identified [46]. The internal consistency investigates whether the items of the scale show a consistent or similar response, given by a good mutual correlation, which was observed in the FaceSed.

Differently from internal consistency, which confirmed the intercorrelation of the items, item-total correlation tests the homogeneity of the scale based on the correlation of a particular item with the scale as a whole, omitting the target item [46]. Therefore, the contribution of each item is analysed independently from the other items. The items of the FaceSed contributed homogeneously to the sum of the scale [56], because 0.3 is the minimum correlation for an item to have a significant role on the scale [46]. Values above 0.7 indicate that the item only repeats the trend of the scale and could be redundant [46]. All items, except orbital opening (0.73), were within this range, showing that orbital opening may be a restatement of other items. This item was maintained in the scale because the item-total correlation was close to the maximum limit and it was within the approval criteria of all other tests.

All items presented sensitivity to identify sedated horses. As for specificity, only the items orbital opening and upper lip relaxation were specific in detecting truthfully non-sedated animals. The items ears and lower lip were not specific, as scores 1 were attributed at baseline, as shown in the frequency distribution data. The lack of specificity of these two items is a limitation of the FaceSed, as it indicates that horses in their natural and relaxed state, present a false sedation characteristic [33].

To define the cut-off value of the FaceSed, the HHAG% was used as a predictive value, where horses with scores > 50% were considered sufficiently sedated [14]. According to the Youden index of the FaceSed, horses scoring > 5 are adequately sedated for standing procedures.

Although the horse grimace scale [31] along with other facial scales [26, 32, 33, 35, 36] were the starting point to develop the FaceSed, it was not feasible only to invert the scores of the horse grimace scale and use the same instrument to assess sedation, because the FaceSed was developed with specific descriptors of sedation in horses of the present study. Pain produces muscle contraction, while sedation produces muscle relaxation, therefore only the neutral point (normal state) is coincident in both conditions and scales, i.e., in the presence of sedation in the FaceSed, the lower and upper lips are relaxed and in the presence of pain they are contracted [31]. Ears are stiff and directed backwards in horses undergoing pain according to the horse grimace scale [31], otherwise they are relaxed and open in horses under sedation (FaceSed). Therefore during the process of content validation, it was necessary to include different descriptors.

With regard to ear position, we observed that in a sedated horse the lateral distance between the tips of the ears is widened, as described by the EquiFACS (Equine Facial Action Coding

System) in relaxed horses, with reduced visibility of the inner ear in the lateral view [54]. The movement [26] and reaction to touch [9] of the ears have already been targets of the investigation to evaluate sedation, however their static positioning has not been well described. The evaluation of the ability of the ears to move, without considering position, may not discriminate between a horse that is sedated and/or in pain. In the case of pain, the distance between the tips of the ears is also increased, but in a backward direction or asymmetry instead [31, 44].

The facial action unit orbital closure described by the EquiFACS in relaxed horses, occurs due to the relaxation of the upper eyelid levator palpebrae superioris muscle [54]. Orbital opening/closure was the only coincident item between the horse grimace scale and FaceSed. It should not be confused with the orbital tightening shown in the horse grimace scale of horses with pain, where there is tension above the eye area [31] due to contraction of the muscles above the eye area. The lower lip relaxation presented low specificity in the FaceSed which may be justified by its occurrence in non-sedated relaxed horses as well [54].

The upper lip relaxation accompanies the vertical stretching of the nostrils, also described in the EquiFACS as a change in the conformation of the edges of the nostrils from a curved to a more elongated shape [54]. However, in EquiFACS this action occurs together with the elevation of the nostril [54, 55], and not the relaxation of the lip as observed in the present study in sedated horses. Inclusion of a specific item for nose changes might be prone to human biases, because when people process facial expressions, noses do not attract attention from people and other animals [77]. The dynamic perception of emotions by human beings are modulated by their main area of interest which starts by analysing the eyes and mouth, for both joy and fear, and so sedation signals of the FaseSed could be missed [77, 78]. Facial scales developed to identify pain or sedation might be susceptible to human biases if they are not performed in a systematic way like EquiFACS [54].

Our study is not free of limitations. One bias that needs to be considered is that all evaluators knew that the objective of the study was to identify sedation and two evaluators were present during the experimental phase. To reduce the expectation bias, evaluators were independent and blinded to the treatments. Furthermore, as previously mentioned, sweating and position of the neck might be apparent in the lateral positioning photos, therefore, at some time-points the sedation depth might be identifiable, which could affect scoring decision making. The first limitation of the study was that the horses were docile, acclimated to the site, experimental handling and face-to-face evaluators. This may have contributed to the false interpretation that the horses were sedated at baseline, therefore reducing specificity values. To overcome this drawback, the reliability of FaceSed should be tested in different scenarios and environments, with various types of handling, and by different people from those the horses are used to, to ensure that the instrument presents the same reliability. A second limitation was that the photographic record was not made from videos, which may not identify the full expression of facial regions acting or relaxing at the moment of evaluation [43]. Thus, it would be advisable to apply the scale through short videos instead of photographs and scale on-site. Either way, there is still the bias of the photographer to determine the moments for picture capture. Furthermore, measuring movement in photographs is difficult and a confounding factor might be the breed differences when accessing photos (i.e. eye wrinkle) [79]. Another limitation of the FaceSed along with other facial scales, is that it would be difficult to apply for dental and ophthalmic interventions.

A final limitation, as mentioned above was that, as in numerical rating scale and HHAG%, the FaceSed was not capable of differentiating tranquilisation from low sedation. Further refinements in facial recognition, which may include not only visual analysis of images but also geometric morphometric approaches, as reported in cats [80] or even, in the future, deep learning tools [81] might be useful to differentiate these stages of responses to drugs.

In conclusion, the FaceSed presents content, criteria, construct validities and adequate intra and interobserver reliability to identify both tranquilisation and sedation in horses when assessed by trained anaesthesiologists. Further studies in clinical and other experimental scenarios and assessment by inexperienced observers may either confirm or not whether facial sedative characteristics evaluated on-site will present similar results. At this stage, FaceSed is a short, easy to apply scale and may be useful in clinical practice and in research purposes. Other main advantages include that it demands a short time to be applied without interaction with the horse.

## Supporting information

**S1 Appendix. Flowchart of the methodology of the data collection of a facial sedation scale in horses.**
(TIF)

**S2 Appendix. Guidelines for evaluation of a facial sedation scale in horses.**
(DOCX)

**S1 Data.**
(XLSX)

## Acknowledgments

We thank the colleagues who kindly participated in the EquiSed content validation: Marilda Onghero Taffarel, Flávia Augusta de Oliveira, and Carlize Lopes, and Altamiro Rosam for the care and dedication to the animals.

## Author Contributions

**Conceptualization:** Alice Rodrigues de Oliveira, Miguel Gozalo-Marcilla, Pedro Henrique Esteves Trindade, Stelio Pacca Loureiro Luna.

**Data curation:** Alice Rodrigues de Oliveira, Miguel Gozalo-Marcilla, Simone Katja Ringer, Stijn Schauvliege, Mariana Werneck Fonseca, José Nicolau Prospero Puoli Filho.

**Formal analysis:** Alice Rodrigues de Oliveira, Pedro Henrique Esteves Trindade.

**Funding acquisition:** Alice Rodrigues de Oliveira, Stelio Pacca Loureiro Luna.

**Investigation:** Alice Rodrigues de Oliveira, Miguel Gozalo-Marcilla, Simone Katja Ringer, Stijn Schauvliege, Mariana Werneck Fonseca.

**Methodology:** Alice Rodrigues de Oliveira, Miguel Gozalo-Marcilla, Mariana Werneck Fonseca, Pedro Henrique Esteves Trindade, Stelio Pacca Loureiro Luna.

**Project administration:** Alice Rodrigues de Oliveira, Miguel Gozalo-Marcilla, José Nicolau Prospero Puoli Filho, Stelio Pacca Loureiro Luna.

**Software:** Pedro Henrique Esteves Trindade.

**Supervision:** Miguel Gozalo-Marcilla, Stelio Pacca Loureiro Luna.

**Visualization:** Miguel Gozalo-Marcilla.

**Writing – original draft:** Alice Rodrigues de Oliveira, Miguel Gozalo-Marcilla, Simone Katja Ringer, Stijn Schauvliege, Stelio Pacca Loureiro Luna.

**Writing – review & editing:** Alice Rodrigues de Oliveira, Miguel Gozalo-Marcilla, Stelio Pacca Loureiro Luna.

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
