## [Decision Letter · Decision Letter 0]

1 Dec 2020

PONE-D-20-32183

Development and validation of the facial scale (facesed) to evaluate sedation in horses

PLOS ONE

Dear Dr. Luna,

Thank you for submitting your manuscript to PLOS ONE. After careful consideration, we feel that it has merit but does not fully meet PLOS ONE’s publication criteria as it currently stands. Therefore, we invite you to submit a revised version of the manuscript that addresses the points raised during the review process.

We look forward to receiving your revised manuscript.

Kind regards,

Chang-Qing Gao

Academic Editor

PLOS ONE

Journal Requirements:

2. We noted in your submission details that a portion of your manuscript may have been presented or published elsewhere:

'The only already published data was the measurement of head height above the ground (HHAG) collected in situ. However, in the original study, this data is expressed in centimeters and in this study data is reported in percentage. This information has been included in the methods section. The same data (HHAG%) is under submission in another article about a behavioural sedation scale (EquiSed).

The inclusion of HHAG data in this manuscript does not imply in dual publication because data was used only for comparison against the proposed scale.'

Please clarify whether this  publication was peer-reviewed and formally published.

If this work was previously peer-reviewed and published, in the cover letter please provide the reason that this work does not constitute dual publication and should be included in the current manuscript.

Reviewers' comments:

Reviewer's Responses to Questions

**Comments to the Author**

1. Is the manuscript technically sound, and do the data support the conclusions?

Reviewer #1: Partly

Reviewer #2: Yes

2. Has the statistical analysis been performed appropriately and rigorously? 

Reviewer #1: Yes

Reviewer #2: Yes

3. Have the authors made all data underlying the findings in their manuscript fully available?

Reviewer #1: Yes

Reviewer #2: Yes

4. Is the manuscript presented in an intelligible fashion and written in standard English?

Reviewer #1: Yes

Reviewer #2: Yes

5. Review Comments to the Author

Reviewer #1: Interesting study done as part of/in tandem with another study on anaesthesia, where the authors develop and validate a scale for sedation levels in horses through facial cues. The work follows previous publications on the Horse Grimace Scale and combines this with a couple other works to build the FaceSed. The work done to validate the scale is comprehensive, but I think several clarifications and more information are needed at this point. I also think the authors need to better justify the need for this scale as opposed to just using the HGS and invert the scores (in which this study could simply present an application of HGS for sedation contexts).

Line 16: Sorry, maybe I'm missing something, but I don't understand the author contributions marked with "&". There is only one author marked with "&", so contributed equally to whom?

Line 51: I would suggest adding a short paragraph on the different sedation agents usually used here for horses and why, and then in the methods justify based on this why the authors chose the sedation agents for this study. (I am expecting these to be for example the most common ones).

Line 53-57: Please expand on each of these scales, including what they are, how they measure sedation, differences between them, and why, in the author's opinion they are more or less objective. Also, mention briefly if they were used after development (cite all studies if not many were done). This is the state of the art of this introduction, so needs more information as not everyone might be familiar with all the scales published till date. Furthermore, this will set the scene to then explain why the current new method is better than all of these past ones.

Line 54: I guess this will be replied once there is more information about each scale, but is HHAG a scale or a one item measurement? The references are for studies, I assume, applying this measure. But the reference here should be only the publication that developed the scale. So if I want to use this scale, which publication was the pioneer in using this measurement? The subsequent studies can then be cited in applications of this HHAG tool (and the same goes for the other scales).

Line 58-62: why were these scales deemed convenient? In what way? Also, please add which scales are used for clinical or experimental purposes. The name of each scale, whenever mentioned, should be in the text, so the reader doesn't need to go back to the references everytime.

Line 61-62: Do the authors mean here that in order to use some scales, the handler needs to administer certain visual or auditory stimuli to the anaesthetised animal? If so, please clarify this idea, as it seems a bit vague at the moment and give examples of what kind of visual/auditory stimuli are part of the scales. Also, if I understood this correctly, there is another potential problem other than the ones cited by the author already, as delivery of these stimuli (e.g. intensity, quality) will probably vary a lot with the handler, which can create a confounding variable. For example, a lower intensity stimuli, means a lower response and assumed higher sedation, when this might not be true.

Line 63: The content related to facial expressions within each of the scales mentioned above needs to be expanded. Please include, which scales contain facial expression attributes, and give examples for each scale of these facial attributes.

Line 63-64: the second part of this sentence is a bit confusing for me. These scales were developed for the sole purpose of identifying sedation, so how can they not be relevant? I am guessing by the next sentence, the authors mean validation, consistency, reliability, etc., so perhaps the word relevance is better replaced by something else, like psychometric attributes or something similar? The authors vaguely pass the main important idea here, which is that anyone can develop any scale, but the quality of it as a tool for measuring a construct has to be assessed after development. This needs to be made clearer.

Line 63: I know the authors intend to develop a tool, but more background on the tool development is needed. Perhaps around here, the authors need to explain why the face and facial expressions are relevant for 1) pain and thus 2) evaluating sedation, but also 3) for horses. This needs to be argued well, since some scales include facial attributes and others not. There is also the issue of horses being a prey species and hence, facial signs of pain might not be present (adaptively, they would not be present). What is so special about horse faces (and faces in general) that makes them good indicators for sedation scales? I would suggest here maybe mention some human literature on facial expressions, and/or what we know about horse making use of their facial expressions in general (there are some studies on this). This can then be linked with for example, the HGS that is already mentioned on line 66.

Line 66: It is great to have these examples, but please mention which ones are being cited. Also, while the examples are important, the reason why a scale needs validation is not because others validated their scales before. So please add why it is important to validate a scale.

Line 69-70: why are they important? Please expand.

Line 74: what criteria? please mention them again. As these are the goals of this study, it has to be clear what the authors propose on doing. The prediction is not really a prediction at the moment. It needs to actually predict what the authors think it's going to measure in the different phases of the experiment, and be justified as well based either on lit or logical statements (if lit is not available). Also, the "why" of this study needs to be added to the goal.

Line 93: Was phase I and phase II done in different days? The same individual was thus subjected to all the treatments, i.e. detomidine, detomidine+methadone, acepromazine, in different days, correct? Please make this clear here.

Also, make it clear that what the current manuscript is reporting, is only phase II, and phase I is described only for understanding of phase II. This is important in order to not plagiarise the work published in phase I.

Line 110: how was the baseline decided?

Line 113: What does in situ registration means?

Line 117: I don't think the links for this and the power sample above need to be in text, they can be transformed as a reference - maybe check PLOS ONE referencing guide or a style book for proper in-text citation of websites.

Line 128-131: I'm assuming this was done in phase I but is part of the study reported in this manuscript, correct? This needs to be made clearer. The authors can simply add something like "For the previous study...", "For this study...". Again, it is important to understand what belongs to the study here reported and the study previously published.

Line 130: Why these time points? Is there previous evidence in the literature indicating these time points as ideal sampling?

Line 134: I was surprised here to see that phase I and II had almost a year in between. This information should come before phase I description. Why was there such a long time interval? This means that all the horses were a year older (or just about).

Line 137: Please include what was exactly measured here. Was there a baseline point, HHAG, sedation evaluations and pictures taken in 6 time points, from 5 till 120min? Why here there were less time points?

Line 143: I am a bit confused here, what is the main study here referring to? The previously published one? I would suggest the authors find a way of better distinguishing between these two studies (e.g. give it a name) and then refer to it consistently and clearly define what was in one study and what is in the current study. As a reviewer, I can't really comment on a previous published study in the same way I should comment on the present work, but there are several moments I am not sure which one is which. I think it's great to take advantage of previous studies to collect data, but the reporting of these studies should be carefully done. As I mentioned before, there are auto-plagiarism concerns when publishing twice the same work, and also it can give the impression this is somehow going towards the practice of salami slicing (see for example: https://doi.org/10.1038/nmat1305 or https://doi.org/10.1007/s10459-019-09876-7). Both in the introduction (goals) and methods, throughout the text, it needs to be perfectly clear what was the goal of the previous published study, what is the goal of this study, how do they differ, and why are they being published in separate articles.

Line 152: I am unsure what reference 28 is supposed to mean here, since it refers to a conference presentation...? First, if the reference is correct, something like "personal communication" should be added, since this is not a peer-reviewed published publication and it is not accessible. In fact, according to PLOS ONE guidelines, this should not even be cited:

Do not cite the following sources in the reference list:

• Personal communications (these should be supported by a letter from the relevant authors but not included in the reference list)

If there are more cases of conference presentations, please correct them.

Second, the "degree of sedation" evaluation criteria needs to be explained (either here or in supplementary materials).

Line 155: Why wasn't the end of sedation also determined through plasmatic concentration?

Line 156-161: The concentration of acepromazine was not measured, correct? If so, the authors shouldn't mention "reduction in concentration", since this was not measured. I am also a bit unsure about the way the justification for the time points is being presented. Obviously, the time points were chosen a priori since they are so uniform (i.e. every half hour), so unless there are studies showing that this is indeed such a regular action of the sedation agents, this needs to be corrected. It is important to either explain this differently or correct the justification for each time point, as at the moment it incurs in circular reasoning, i.e. the authors chose 60m, 90m, 120m a priori, then make the degree of sedation scores fit these points and use it to justify why these points were chosen. In order to correct this, either simply assume these time points were chosen for convenience or clearly explain how the degree of sedation scores were calculated (it is ok to go in supplem. materials if needed) and how much they had to increase/decrease in each point. And particularly, carefully justify why it just so happens to be such convenient time points for peak and so on.

Line 164: why is it relevant the 3 other coders are from different institutions and hold this diploma for scoring of facial expressions? All evaluators are authors of this manuscript, so is it right to assume all evaluators were aware of the goal of the coding? Also, was any of the evaluators present on site during the sedation experiments? If so, please add this information.

Line 170: Which author randomised and selected the photos? If one of the evaluators also did this, then please clearly state that this evaluator was not blind to the treatments. If not, please add who did this step.

Line 175: How many rounds of coding were needed to obtain 80%? Was this ICC test? If not, which correlation type was used? And what was the intra-observer reliability for the training? (I'm assuming that the evaluations done one month apart were for this - if not, please explain why that was done). Was the intra-observer taken into account before moving to the test coding?

Line 178-180: This is really not relevant information. What we need to know is what information was given to the evaluators with each photo pair (if any).

Line 181: Can these instructions be added as supplementary materials please?

Line 183: Is the NRS used here published anywhere? If so, please provide references. If not, please add the scale in supplem. materials. In any case, briefly describe what this scale is and how is it applied. Were all the evaluators very familiar with this scale?

Line 188: So the FaceSed is based on the HGS? Please make this clear (instead of vaguely indicating a reference that makes the reader go down to the reference list to understand what is being said), as it is an important methodological step. Also make it clear that eye tightening, strained mouth and stiff ears are indicators of pain in the HGS. Finally, please add more specific descriptions of what these terms mean exactly (according to the HGS).

Line 189: When I read this, the first though in my mind was: why do we need the FaceSed then, since we can just invert the scores from the HGS? Maybe rephrase this sentence or explain better how is it any different from the HGS. If FaceSed is basically scoring the relaxation of muscles from an apex point to a neutral point, then we don't really need it, as muscles do not contract and relax in different ways. How does FaceSed differs from an inverted HGS? Is it because it is combined with EquiFACS and other scales? This is a crucial point to defend in this manuscript, because it might be pointless to create yet another scale, if we could just use the HGS. (if this was the case, then the manuscript would then have to be reframed entirely in order to make clear that this is a validation/adaption of the HGS for a sedation situation and not a creation of a totally new scale).

Line 190-192: I don't quite understand this part of the sentence... photos from the other studies were used to describe sedation scores how? Also, how did untreated pictures of horses contribute to describe sedation scores? This needs to be rephrased as it is a bit confusing and vague. Also, the study cited after EquiFACS is not a study on untreated horses per se, but it is instead a publication detailing the development of a FACS tool to measure muscle-based facial movements in horses (in any situation). Furthermore, the studies cited in 20,21,30, are development of scales, so please name each scale from where the information was extracted. This should be corrected.

Line 192: If FaceSed is derived from a combination of HGS and EquiFACS, was any of the initial developers certified in EquiFACS? Please be aware that only certified coders in EquiFACS can use it. If yes, please add the name of which authors are EquiFACS certified, and if not please clearly state this. However, please be aware that if none of the authors is EquiFACS certified and still is using it, this is going against the guidelines for appropriate use of this tool. Please see https://www.animalfacs.com/equifacs, for more information. I would in fact, strongly advise the mention to EquiFACS to be deleted as a part of the methods for the FaceSed development if none of the authors has the certification. This can then be discussed as a limitation or a future direction in the discussion.

Line 195: How was the relevance attributed? Was it purely based on the subjective opinion of each evaluator, i.e. educated guesses? If so, please state this clearly, if there were basic criteria, please state them instead. If this was purely based on the subjective opinion of each evaluator, it would be interesting to disclose the reasoning for including/excluding items, or at least examples. heavily relying on "expert opinion" to develop an objective tool is always problematic since it often incurs in confirmatory biases. For example, veterinary anaesthesiologists might have certain impressions/ideas about what happens in the horse's face during anaesthesia, these ideas are then used to decide how to evaluate which items matter for evaluating horse's faces during anaesthesia, and the same ideas are then used to build an "objective" scale. This is particularly important as facial expressions are very subtle and hard to detect by people not trained to detect these facial behaviours. This is even harder to do in pictures, as there's considerably less information to decide if a movement is present of not.

Line 205: Right, so this should have been described above (see my previous comment).

Line 226: Please briefly explain what concurrent criteria validation is. The same for construct validity and the other validation measures.

Line 271: Please add versions of each software.

Line 427-430: Please explain why there is this discrepancy.

Line 440: But these were not apparent in the pictures, so how can it have influenced the evaluators?

Line 444-446: But this will assume the first scale is of high quality in all measurement components, plus it assumes there is already an existing scale. So I am not sure this is a "need" per se, instead it probably is more a term of comparison with previous work.

Lines 458-460: I didn't really understand what the authors are trying to say here, please rephrase.

Line 467: In other words, all items correlate with each other? Please explain what this means in practice.

Line 474: Does insufflate here means increase?

One thing I am not clear is if all the analysis undertaken here for the FaceSed assume the item for eyes is independent from the mouth or does assume they all are dependent? Maybe this could be discussed around line 478.

Line 494: FACS systems do not describe "characteristics", they describe Action Units. This distinction is important because the facial movements are anatomically-based. Also the name of the muscle is not "elevator", but "Levator palpebrae superioris".

Line 499-503: I was wondering till this paragraph why the authors ignored the nose area for horses. I assumed it simply did not present any movement during pain/sedation. But here the authors actually mention an important cue for sedation on the nose. I think it is important to discuss why the nose is not part of the scale. (there's good evidence to argue that this might be due to human biases and the way people process facial expressions in general, where noses do not attract attention, both in people and other animals, and so pain/sedation cues here might probably be missed. See for example: https://doi.org/10.1007/s10071-020-01348-5 and https://doi.org/10.1016/j.anbehav.2009.11.003). I think this point should be discussed regarding the nose or other facial regions not included in the scale, but also as a general limitation of these types of scales. I understand the authors looked at different instruments published and combined them, but even so, if this is not done in a systematic way, there will always be human biases (both from hardwired facial processing mechanisms, i.e. we don't tend to look at noses so much, and from expert/confirmatory bias, i.e. experts assume the important cues are in certain regions and do not attend to other regions).

Line 509-511: This is a very good point, but I would add that other than being extremely hard to capture the full expression (I am assuming here full means maximum intensity or with all facial regions acting/relaxing), it is up to the photographer to determine the moments of capture. Furthermore, measuring movement in photographs is very difficult, as it doesn't account for individual differences. See for example: https://doi.org/10.3389/fvets.2019.00154.

Line 514: Can the authors expand on an example of this if already exists in horses, or in other animals if published? The reference provided is not appropriate as it is not a computational approach (the landmarks are manually defined in each picture). A better example of what the authors are trying to say is perhaps this: https://doi.org/10.1016/j.conb.2019.10.008 (DeepLabCut Model Zoo has a module for primate faces with 45 markers that track movement).

Does FaceSed need training or specific backgrounds (e.g. only anaesthesiologists, only people very familiar with horses) to be used or anyone can use it? Please add that to your final remarks.

The ears pictures all have zero above them.

Reviewer #2: dear authors,

you have performed a very nice study that will add to the knowledge on objective assessment of quality of sedation in horses. The reviewer only has some minor issues that should be addressed, among them there are some questions about the statistics you have used and about the way criteria validation has been performed:

abstract:

line 23: "measuring" instead of "measured"

line 25: "performed" instead of "perfomed"

line 26: instead of "associated" you could better use the word "combined"

line 32: you state that intra- and inter observer reliability were good to very good, but the reader does not know at this stage what the values 0.74-0.94 are? (ICC, Crohnbach's alpha??)

line 36: you state in this line that the scale was unidimensional according to principal component analysis, why? maybe some explanation over here?

line 38: you state in this line that intem-total correlation was adequate, although te range of values is 0.3-0.73. It is not clear again what values these are (ICC?), but the lower range of 0.3 does not seem to be high enough for adequate?

introduction:

line 49: you mention standing horses, what about the role of sedative drugs in premedication for general anaesthesia?

line 64: instead of "sedation" you could maybe better use "depth and quality of sedation"?

materials and methods:

line 80-81 and line 95: what do you mean with "this study was opportunistic of another study"?

line 90: instead of "has begun" you could better use "began" or "would begin"

line 96-101: the reviewer thinks you should mention over here what the calculated sample size was?

phase I:

line 107: instead of "data" you should use the word "parameter"

line 113-114: what do you mean with "in situ registration"?

line 116: how was the treatment randomized?

line 117: it is not clear what the website you mention over here should tell the reader?

line 118: "Those" probably reflects to the treatment, but this is not grammatically sound

Selection of photos and evaluations

line 151: "that" instead of "who"

line 151: "plasma" instead of "plasmatic"

line 151-152: what do you mean with "corroborated by the degree of sedation recorded in situ"??

line 161: isn't 120 minutes after ACP administration a bit early for the higher dosage of ACP to determine end of tranquilisation?

line 162: "did not communicate with each other" instead of "between them"

line 174-176: why did you choose this 80% as the correlation that was needed to start the main evaluations?

line 183-184: you mention over here that the observers were asked to score the NRS first and after that the FaceSed. Why did you choose for this order? The main research question was about FaceSed and not about the NRS scoring, right? Might this previous NRS scoring have influenced the following FaceSed scoring? this could be a source of bias that is incorporated in the study design.

It would have ben much better if the NRS scores and FaceSed scores would have been disconnected from each other. Another problem is that you analyze the correlation between the NRS and the Face Sed, but because these were scored immediately after another, these observations were not independent and the correlation is therefore not valid.

development of the Facesed scale

line 193: "experience in sedation" instead of "experience of sedation

Statistical analyses

line 204: "...who attributed to the importance...."

line 210: what do you mean with "the following analysis"?

line 211: what do you mean with in-situ?

line 230: for concurrent criteria validation, the reviewer thinks that it is not the most logical choice to compare the FaceSed with the NRS, since this scale also holds some subjectivity and since these were assessed and taken immediately after another, they aare not independent. Therefore, it would be better to compare FaceSed only with the HHAG%, since this is a completely objective and independent parameter.

line 241-242: this sentence is not grammatically sound, what do you want to explain with this sentence?

line 243: I think you would need to explain what the eigen values and variance mean. These are not necessarily parameters that all readers are familiair with.

line 255: the range of acceptable values for the item-total correlation of 0.3 to 0.7 seems to have a very low lower margin for acceptable Spearman correlation? A correlation of 0.3 would seem to be very low according and not acceptable to the reviewer?

line 268: I would change the text into "...the frequency of sedation scores that were assigned by the evaluators...."

line 272: you could add "statistical significance was accepted at P<0.05?

Results

line 295: the range of repeatability of the FaceSed sum of 0.74-0.94 seems to be the combined range of all 4 observers taken together. This should be mentioned like this in this line.

line 298: instead of "steps" it would be better to use the term "observations"

line 372: like mentioned earlier, a correlation of 0.3 does not seems to be acceptable according to the reviewer.

line 373: in this line you state that all parameters showed an item-total correlation that was acceptable (although I would doubt this with a correlation of 0.3), except for the item eyes. What was the item-total correlation for this parameter?

line 378-379: The Crohnbach's alpha was used as a measure of internal consistency. With what was the FaceSed score compared to determine this value?

Discussion:

line 409: "low degree of sedation" instead of "low sedation"

line 427-428: could you maybe discuss this difference , for instance due to the technically more difficult task of observing ridden horses?

line 432: "the biases that could affect.."

line 436-437: you are right that scoring of the FaceSed did not bias scoring of the NRS, but the other way around, there might be a reason for bias. The fact that the NRS was first scored and immediately after that the FaceSed, might have influenced the reproducibility and repeatability of FaceSed and the correlation between FaceSed and NRS. This is what you mention in line 446, however, this high correlation might be due to how the scoring was performed.

line 448: this is not how you describe it earlier in your materials and methods: there, you say that for criterion validity, the FaceSed is compared to the NRS and the HHAG%.

line 461: instead of "....tranquilisation and low and high sedation.." you could better formulate "...tranquilisation with ACP from low and high sedation intensities with detomidine (with or without methadone)". Maybe could you also hypothesize about a possible reason for this?

line 484: in this paragraph you discuss sensitivity and specificity. Was it possible to determine vcut-off values that optimally discriminate between non-sedated and sedated horses?

line 494-497: you describe the difference betwen orbital closure due to being relaxed compared to due to being in pain. This lack of discriminatory power on this parameter could also lead to a false positive score, could you underline this better maybe?

line 513: instead of "of", you could better use "..tranquilisation from..."

6. PLOS authors have the option to publish the peer review history of their article (what does this mean?). If published, this will include your full peer review and any attached files.

Reviewer #1: No

Reviewer #2: No

---

## [Author Response · Author response to Decision Letter 0]

22 Mar 2021

PONE-D-20-32183

Development and validation of the facial scale (FaceSed) to evaluate sedation in horses

PLOS ONE

Dear Dr. Luna,

Thank you for submitting your manuscript to PLOS ONE. After careful consideration, we feel that it has merit but does not fully meet PLOS ONE’s publication criteria as it currently stands. Therefore, we invite you to submit a revised version of the manuscript that addresses the points raised during the review process.

We look forward to receiving your revised manuscript.

Kind regards,

Chang-Qing Gao

Academic Editor

PLOS ONE

Journal Requirements:

2. We noted in your submission details that a portion of your manuscript may have been presented or published elsewhere:

Answer: 'The only data that have been previously published are the measurement of head height above the ground (HHAG) collected on site. However, in the original study(Gozalo-Marcilla et al. Sedative and antinociceptive effects of different detomidine constant rate infusions, with or without methadone in standing horses. Equine Vet J. 2018;51: 530–536. doi:10.1111/evj.13054), these data are expressed in centimeters and in the present study in percentages. This information has been included in the methods section. The same data (HHAG%) have been published in another article about a behavioural sedation scale (EquiSed) (Frontiers in Veterinary Medicinedoi: 10.3389/fvets.2021.611729).

To our knowledge the inclusion of HHAG data in this manuscript does not imply in dual publication because the data were used only for comparison against the proposed scale.

Please clarify whether this publication was peer-reviewed and formally published.

Answer: yes, both publications have been peer reviewed. One has been formally published (Gozalo-Marcilla et al. Sedative and antinociceptive effects of different detomidine constant rate infusions, with or without methadone in standing horses. Equine Vet J. 2018;51: 530–536. doi:10.1111/evj.13054) and the other is recently published (Oliveira et al. Development, Validation, and Reliability of a Sedation Scale in Horses (EquiSed). Frontiers in Veterinary Medicinedoi: 10.3389/fvets.2021.611729) 

If this work was previously peer-reviewed and published, in the cover letter please provide the reason that this work does not constitute dual publication and should be included in the current manuscript.

Answer: To our knowledge the specific data of HHAG are not considered dual publication because they are used only for comparison with the Facial scale. This is the only objective and standard method to identify sedation in horses that can be used to compare with a new sedation method being developed (FaceSed) (line 497– Table 4).

Answer: Included line 762

Reviewers' comments:

Reviewer's Responses to Questions

Comments to the Author

1. Is the manuscript technically sound, and do the data support the conclusions?

Reviewer #1: Partly

Answer: Dear Reviewer, we hope to have satisfied this item after this revision. If not, please let us know what further requirements are necessary.

Reviewer #2: Yes

2. Has the statistical analysis been performed appropriately and rigorously?

Reviewer #1: Yes

Reviewer #2: Yes

3. Have the authors made all data underlying the findings in their manuscript fully available?

Reviewer #1: Yes

Reviewer #2: Yes

4. Is the manuscript presented in an intelligible fashion and written in standard English?

Reviewer #1: Yes

Reviewer #2: Yes

5. Review Comments to the Author

Reviewer #1: Interesting study done as part of/in tandem with another study on anaesthesia, where the authors develop and validate a scale for sedation levels in horses through facial cues. The work follows previous publications on the Horse Grimace Scale and combines this with a couple other works to build the FaceSed. The work done to validate the scale is comprehensive, but I think several clarifications and more information are needed at this point. I also think the authors need to better justify the need for this scale as opposed to just using the HGS and invert the scores (in which this study could simply present an application of HGS for sedation contexts).

Answer: Dear Reviewer

The authors appreciate your time and effort spent reviewing this manuscript, and thank you very much for your comments. All corrections have been performed according to the Reviewers' suggestions, and each comment responded to separately. Changes are highlighted according to the Journal's recommendations. We hope that after these corrections, you consider the manuscript suitable for publication, but we are happy to answer any further questions.

Please find below the justification regarding the need for this scale as opposed to simply using the HGS and inverting the scores.

Line 16: Sorry, maybe I'm missing something, but I don't understand the author contributions marked with "&". There is only one author marked with "&", so contributed equally to whom?

Answer: Excluded.

Line 51: I would suggest adding a short paragraph on the different sedation agents usually used here for horses and why, and then in the methods justify based on this why the authors chose the sedation agents for this study. (I am expecting these to be for example the most common ones).

Answer: Included – introduction (lines 52 - 58) and methods (lines 171 -175)

Line 53-57: Please expand on each of these scales, including what they are, how they measure sedation, differences between them, and why, in the author's opinion they are more or less objective. Also, mention briefly if they were used after development (cite all studies if not many were done). This is the state of the art of this introduction, so needs more information as not everyone might be familiar with all the scales published till date. Furthermore, this will set the scene to then explain why the current new method is better than all of these past ones.

Answer: The scales have been described in detail along with the differences between them (lines 59-83).

Line 54: I guess this will be replied once there is more information about each scale, but is HHAG a scale or a one item measurement? The references are for studies, I assume, applying this measure. But the reference here should be only the publication that developed the scale. So if I want to use this scale, which publication was the pioneer in using this measurement? The subsequent studies can then be cited in applications of this HHAG tool (and the same goes for the other scales).

Answer: More information regarding the scales has been included (lines 59 – 83) as well as the pioneer references. HHAG is only one item measurement and this has been clarified.

Line 58-62: why were these scales deemed convenient? In what way? Also, please add which scales are used for clinical or experimental purposes. The name of each scale, whenever mentioned, should be in the text, so the reader doesn't need to go back to the references everytime.

Answer: A more detailed description has been incorporated, including the limitations (lines 84 – 94).

Line 61-62: Do the authors mean here that in order to use some scales, the handler needs to administer certain visual or auditory stimuli to the anaesthetised animal? If so, please clarify this idea, as it seems a bit vague at the moment and give examples of what kind of visual/auditory stimuli are part of the scales. Also, if I understood this correctly, there is another potential problem other than the ones cited by the author already, as delivery of these stimuli (e.g. intensity, quality) will probably vary a lot with the handler, which can create a confounding variable. For example, a lower intensity stimuli, means a lower response and assumed higher sedation, when this might not be true.

Answer: Thanks for pointing that out. We believe the description of the stimuli performed in other studies has now been clarified along with the different scales used (lines 84 – 94). The variation in the delivery of these stimuli according to the handler has also been included as a limitation of studies assessing sedation scales (lines 92-94).

Line 63: The content related to facial expressions within each of the scales mentioned above needs to be expanded. Please include, which scales contain facial expression attributes, and give examples for each scale of these facial attributes.

Answer: Included lines 95 – 106. 

Line 63-64: the second part of this sentence is a bit confusing for me. These scales were developed for the sole purpose of identifying sedation, so how can they not be relevant? I am guessing by the next sentence, the authors mean validation, consistency, reliability, etc., so perhaps the word relevance is better replaced by something else, like psychometric attributes or something similar? The authors vaguely pass the main important idea here, which is that anyone can develop any scale, but the quality of it as a tool for measuring a construct has to be assessed after development. This needs to be made clearer.

Answer: Thanks for pointing that out. The whole paragraph has been amended and details added (lines 95 - 106 and 121 – 122).

Line 63: I know the authors intend to develop a tool, but more background on the tool development is needed. Perhaps around here, the authors need to explain why the face and facial expressions are relevant for 1) pain and thus 2) evaluating sedation, but also 3) for horses. This needs to be argued well, since some scales include facial attributes and others not. There is also the issue of horses being a prey species and hence, facial signs of pain might not be present (adaptively, they would not be present). What is so special about horse faces (and faces in general) that makes them good indicators for sedation scales? I would suggest here maybe mention some human literature on facial expressions, and/or what we know about horse making use of their facial expressions in general (there are some studies on this). This can then be linked with for example, the HGS that is already mentioned on line 66.

Answer: Background about facial expression and its relevance in horses has been included, lines 95 -106.

Line 66: It is great to have these examples, but please mention which ones are being cited. Also, while the examples are important, the reason why a scale needs validation is not because others validated their scales before. So please add why it is important to validate a scale.

Answer: The scales have been mentioned, lines 109 – 110 and the reason behind scale validation has been included, lines 107-109.

Line 69-70: why are they important? Please expand.

Answer: Included lines 113-120.

Line 74: what criteria? please mention them again. As these are the goals of this study, it has to be clear what the authors propose on doing. The prediction is not really a prediction at the moment. It needs to actually predict what the authors think it's going to measure in the different phases of the experiment, and be justified as well based either on lit or logical statements (if lit is not available). Also, the "why" of this study needs to be added to the goal.

Answer: The criteria and predictions have been included in lines 121 – 132. The reason why the study was performed has been described in lines 126 – 139.

Line 93: Was phase I and phase II done in different days? The same individual was thus subjected to all the treatments, i.e. detomidine, detomidine+methadone, acepromazine, in different days, correct? Please make this clear here.

Answer: Phases I and II were performed on different days with the same horses, eleven months apart. This information has been included, lines 160 – 164.

Also, make it clear that what the current manuscript is reporting, is only phase II, and phase I is described only for understanding of phase II. This is important in order to not plagiarise the work published in phase I.

Answer: This manuscript is reporting Phases I and II and the only data that have been replicated in this manuscript in common with the previous study are values of height of the head above the ground. Included, lines 179 – 181.

Line 110: how was the baseline decided?

Answer: The baseline was the measurement of the head height above the ground when the horse was unsedated, before the administration of the treatments. This has been included, lines 184 – 187.

Line 113: What does in situ registration means?

Answer: Excluded because it is not relevant (in situ = at the experimental moment).

Line 117: I don't think the links for this and the power sample above need to be in text, they can be transformed as a reference - maybe check PLOS ONE referencing guide or a style book for proper in-text citation of websites.

Answer: The link has been Included as a reference.

Line 128-131: I'm assuming this was done in phase I but is part of the study reported in this manuscript, correct? This needs to be made clearer. The authors can simply add something like "For the previous study...", "For this study...". Again, it is important to understand what belongs to the study here reported and the study previously published.

Answer: We believe this is now better explained, line 206 and line 230. Please let us know if further clarification is required.

Line 130: Why these time points? Is there previous evidence in the literature indicating these time points as ideal sampling?

Answer: These time points were selected according to previous studies with similar methodology and pharmacokinetic data. References have been included in lines 209 - 211.

Line 134: I was surprised here to see that phase I and II had almost a year in between. This information should come before phase I description. Why was there such a long time interval? This means that all the horses were a year older (or just about).

Answer: This information has been included before Phase I, line 161. We decided to include the acepromazine groups after Phase I to assess the validity of the FaceSed not only for sedation but also for tranquilization. Unfortunately, the facilities, horses, and authors were not available to perform the study earlier. This has been included in the discussion (line 591 - 595).

Line 137: Please include what was exactly measured here. Was there a baseline point, HHAG, sedation evaluations and pictures taken in 6 time points, from 5 till 120min? Why here there were less time points?

Answer: More details have been included. The horses were only maintained for 120 minutes because during this phase there was no other concomitant study, therefore it was not necessary to maintain the horses in the stall for longer than 120 minutes as in phase I. Lines 221 – 225.

Line 143: I am a bit confused here, what is the main study here referring to? The previously published one? I would suggest the authors find a way of better distinguishing between these two studies (e.g. give it a name) and then refer to it consistently and clearly define what was in one study and what is in the current study. As a reviewer, I can't really comment on a previous published study in the same way I should comment on the present work, but there are several moments I am not sure which one is which. I think it's great to take advantage of previous studies to collect data, but the reporting of these studies should be carefully done. As I mentioned before, there are auto-plagiarism concerns when publishing twice the same work, and also it can give the impression this is somehow going towards the practice of salami slicing (see for example: https://doi.org/10.1038/nmat1305 or https://doi.org/10.1007/s10459-019-09876-7). Both in the introduction (goals) and methods, throughout the text, it needs to be perfectly clear what was the goal of the previous published study, what is the goal of this study, how do they differ, and why are they being published in separate articles.

Answer: Corrections have been performed throughout the manuscript to differentiate data collected from the previous and present studies. We had to split these studies because they had completely different objectives and the amount of data would be excessive for only one publication (line 144-149). Thanks for addressing these points, your comments were very useful to improve the clarity and quality of the present manuscript.

Line 152: I am unsure what reference 28 is supposed to mean here, since it refers to a conference presentation...? First, if the reference is correct, something like "personal communication" should be added, since this is not a peer-reviewed published publication and it is not accessible. In fact, according to PLOS ONE guidelines, this should not even be cited:

Do not cite the following sources in the reference list:

• Personal communications (these should be supported by a letter from the relevant authors but not included in the reference list)

If there are more cases of conference presentations, please correct them.

Answer: We are sorry for referencing a conference abstract. This has been removed and replaced with the recently published article

Second, the "degree of sedation" evaluation criteria needs to be explained (either here or in supplementary materials).

Answer: Included, lines 238– 240.

Line 155: Why wasn't the end of sedation also determined through plasmatic concentration?

Answer: The end of sedation time-point was determined based on low sedation scores and on the residual plasma concentration according to the previous pharmacokinetic study performed simultaneously. This information has been included, line 244.

Line 156-161: The concentration of acepromazine was not measured, correct? If so, the authors shouldn't mention "reduction in concentration", since this was not measured.

Answer: The authors agree that ”reduction in concentration” was misused. Corrected, line 249-250.

 I am also a bit unsure about the way the justification for the time points is being presented. Obviously, the time points were chosen a priori since they are so uniform (i.e. every half hour), so unless there are studies showing that this is indeed such a regular action of the sedation agents, this needs to be corrected. It is important to either explain this differently or correct the justification for each time point, as at the moment it incurs in circular reasoning, i.e. the authors chose 60m, 90m, 120m a priori, then make the degree of sedation scores fit these points and use it to justify why these points were chosen. In order to correct this, either simply assume these time points were chosen for convenience or clearly explain how the degree of sedation scores were calculated (it is ok to go in supplem. materials if needed) and how much they had to increase/decrease in each point. And particularly, carefully justify why it just so happens to be such convenient time points for peak and so on.

Answer: The reasons behind the selected time points have been better described in lines 245 – 255.

Line 164: why is it relevant the 3 other coders are from different institutions and hold this diploma for scoring of facial expressions? All evaluators are authors of this manuscript, so is it right to assume all evaluators were aware of the goal of the coding? Also, was any of the evaluators present on site during the sedation experiments? If so, please add this information.

Answer: Independent blind evaluators are relevant for validation studies in order to minimize expectation bias. The main author was present during the experimental work and was responsible for selecting the photos. We have included the information that the observers were Diplomate in Veterinary Anaesthesia only to inform readers that the observers have experience in the area, therefore the scale must be tested in the future with inexperienced observers to check its reliability. This information has been included in the manuscript (methods: Lines 261-266 and discussion: lines 722 - 727).

Line 170: Which author randomised and selected the photos? If one of the evaluators also did this, then please clearly state that this evaluator was not blind to the treatments. If not, please add who did this step.

Answer: Included, lines 263 and 269 - 270.

Line 175: How many rounds of coding were needed to obtain 80%? Was this ICC test? If not, which correlation type was used? And what was the intra-observer reliability for the training? (I'm assuming that the evaluations done one month apart were for this - if not, please explain why that was done). Was the intra-observer taken into account before moving to the test coding?

Answer: The evaluators scored the photos only twice, with an interval of one month between viewings (273-278). The intra- and inter-observer reliability was 80% (Spearman correlation). The intra-observer reliability is the comparison between the first and second evaluation for each observer and the inter-observer reliability is the matrix correlation comparison among observers at the second evaluation. Spearman correlation analysis was performed and included (line 273-276), both intra and inter observer reliability were over 0.8.

Line 178-180: This is really not relevant information. What we need to know is what information was given to the evaluators with each photo pair (if any).

Answer: This information has been excluded, and guidelines are available as supplementary information (Guidelines), lines 282 to 285.

Line 181: Can these instructions be added as supplementary materials please?

Answer: Supplementary information has been included and attached to the manuscript, line 282.

Line 183: Is the NRS used here published anywhere? If so, please provide references. If not, please add the scale in supplem. materials. In any case, briefly describe what this scale is and how is it applied. Were all the evaluators very familiar with this scale?

Answer: The NRS used here was published and used for pain assessment – reference included in line 294. It is a simple and intuitive scale. The evaluators were instructed in the guidelines on how to use the scale (supplementary material) before starting the analysis of the photos. This has been included in lines 282 – 285.

Line 188: So the FaceSed is based on the HGS? Please make this clear (instead of vaguely indicating a reference that makes the reader go down to the reference list to understand what is being said), as it is an important methodological step. Also make it clear that eye tightening, strained mouth and stiff ears are indicators of pain in the HGS. Finally, please add more specific descriptions of what these terms mean exactly (according to the HGS).

Answer: A detailed description has been included in lines 288 – 309.

Line 189: When I read this, the first though in my mind was: why do we need the FaceSed then, since we can just invert the scores from the HGS? Maybe rephrase this sentence or explain better how is it any different from the HGS. If FaceSed is basically scoring the relaxation of muscles from an apex point to a neutral point, then we don't really need it, as muscles do not contract and relax in different ways. How does FaceSed differs from an inverted HGS? Is it because it is combined with EquiFACS and other scales? This is a crucial point to defend in this manuscript, because it might be pointless to create yet another scale, if we could just use the HGS. (if this was the case, then the manuscript would then have to be reframed entirely in order to make clear that this is a validation/adaption of the HGS for a sedation situation and not a creation of a totally new scale).

Answer: Although the FaceSed was based on the main facial characteristics described in the HGS as a starting point, it is not possible only to invert the scores of the HGS, because the FaceSed was developed with specific descriptors characteristic of sedation in the horses of the present study (this has been included in lines 684 – 687 of Discussion section). The authors considered the reviewer's suggestions and have rewritten the description of the scale development (lines 288- 309), including the differences between the FaceSed and HGS.

Only the orbital aperture was coincident in both scales. The basic difference between the scales is that, except for orbital aperture, the neutral point is the only coincident score between both scales. Pain produces contraction and sedation produces relaxation, therefore simply inverting the score does not provide the same facial expression. This has been better explained in lines 684 – 694.

Line 190-192: I don't quite understand this part of the sentence... photos from the other studies were used to describe sedation scores how? Also, how did untreated pictures of horses contribute to describe sedation scores? This needs to be rephrased as it is a bit confusing and vague. Also, the study cited after EquiFACS is not a study on untreated horses per se, but it is instead a publication detailing the development of a FACS tool to measure muscle-based facial movements in horses (in any situation). Furthermore, the studies cited in 20,21,30, are development of scales, so please name each scale from where the information was extracted. This should be corrected.

Answer: Thanks for your contribution to the Method description. The methodology of the development of the scale has been completely rephrased. We hope now that it is more comprehensive, as we used other studies to decide which facial action units would be representative to evaluate sedation from the state of normality (lines 684 – 694).

Answer: We only used EquiFACS to identify possible indicators of movement of the facial musculature which would resemble easily identifiable facial units described in other studies of horses under pain and sedation. We did not use or apply EquiFACS in our assessments because none of the authors is EquiFACS certified. I hope this issue is now better described– lines 300-303.

Line 195: How was the relevance attributed? Was it purely based on the subjective opinion of each evaluator, i.e. educated guesses? If so, please state this clearly, if there were basic criteria, please state them instead. If this was purely based on the subjective opinion of each evaluator, it would be interesting to disclose the reasoning for including/excluding items, or at least examples. heavily relying on "expert opinion" to develop an objective tool is always problematic since it often incurs in confirmatory biases. For example, veterinary anaesthesiologists might have certain impressions/ideas about what happens in the horse's face during anaesthesia, these ideas are then used to decide how to evaluate which items matter for evaluating horse's faces during anaesthesia, and the same ideas are then used to build an "objective" scale. This is particularly important as facial expressions are very subtle and hard to detect by people not trained to detect these facial behaviours. This is even harder to do in pictures, as there's considerably less information to decide if a movement is present of not.

Answer: The content validation description was the first item of statistical analysis. It has been moved to before the Statistical analysis and included as supplementary information. Content validation was achieved in three steps explained in the manuscript. Although an “expert” committee is a traditional way to assess content validity, we included two other steps to minimize subjective opinion (Streiner DL, Norman GR, Cairney J. Health measurement scales: A practical guide to their development and use. 5th ed. New York: Oxford University Press; 2015)– lines 310 – 320.

Line 205: Right, so this should have been described above (see my previous comment).

Answer: Included before statistical analysis as suggested.

Line 226: Please briefly explain what concurrent criteria validation is. The same for construct validity and the other validation measures.

Answer: More detailed information has been included. The most common way to test concurrent criteria validation is to correlate the proposal instrument with a gold-standard one. The construct validity or responsiveness consists of changes over time indicative of the presence and absence of the target phenomenon, in our case sedation (lines 339 - 343 and 351 - 353).

Line 271: Please add versions of each software.

Answer: Included in lines 421 – 422. R-Studio Team – 2016.

Line 427-430: Please explain why there is this discrepancy.

Answer: The reason behind the discrepancy between the studies has been included in lines 601 – 603.

Line 440: But these were not apparent in the pictures, so how can it have influenced the evaluators?

Answer: Sweating and low HHAG might be apparent in some pictures. This has been included in the discussion (line 615) and also as a limitation of the study (lines 725 - 727), with information that evaluators were blind to treatments but, because of this limitation, possibly not to the moments.

Line 444-446: But this will assume the first scale is of high quality in all measurement components, plus it assumes there is already an existing scale. So I am not sure this is a "need" per se, instead it probably is more a term of comparison with previous work.

Answer: The paragraph has been rephrased (lines 625 – 630).

Lines 458-460: I didn't really understand what the authors are trying to say here, please rephrase.

Answer: The paragraph has been rephrased (lines 632- 639).

Line 467: In other words, all items correlate with each other? Please explain what this means in practice.

Answer: The practical implication has been included (lines 656-658).

Line 474: Does insufflate here means increase?

Answer: The sentence has been slightly modified to add clarity about the meaning of this value to; ‘showing that the orbital opening (0.73) may be a restatement of other FaceSed items’ (line 671 - 674).

One thing I am not clear is if all the analysis undertaken here for the FaceSed assume the item for eyes is independent from the mouth or does assume they all are dependent? Maybe this could be discussed around line 478.

Answer: a more detailed explanation about item-total correlation has been provided (lines 666 – 671).

Line 494: FACS systems do not describe "characteristics", they describe Action Units. This distinction is important because the facial movements are anatomically-based. Also the name of the muscle is not "elevator", but "Levator palpebrae superioris".

Answer: Corrected to Facial Action Units throughout the manuscript where applicable.

Line 499-503: I was wondering till this paragraph why the authors ignored the nose area for horses. I assumed it simply did not present any movement during pain/sedation. But here the authors actually mention an important cue for sedation on the nose. I think it is important to discuss why the nose is not part of the scale. (there's good evidence to argue that this might be due to human biases and the way people process facial expressions in general, where noses do not attract attention, both in people and other animals, and so pain/sedation cues here might probably be missed. See for example: https://doi.org/10.1007/s10071-020-01348-5 and https://doi.org/10.1016/j.anbehav.2009.11.003). I think this point should be discussed regarding the nose or other facial regions not included in the scale, but also as a general limitation of these types of scales. I understand the authors looked at different instruments published and combined them, but even so, if this is not done in a systematic way, there will always be human biases (both from hardwired facial processing mechanisms, i.e. we don't tend to look at noses so much, and from expert/confirmatory bias, i.e. experts assume the important cues are in certain regions and do not attend to other regions).

Answer: Thank you very much for pointing this out and for the suggestions on the literature. This paragraph has been amended, lines 715 – 719.

Line 509-511: This is a very good point, but I would add that other than being extremely hard to capture the full expression (I am assuming here full means maximum intensity or with all facial regions acting/relaxing), it is up to the photographer to determine the moments of capture. Furthermore, measuring movement in photographs is very difficult, as it doesn't account for individual differences. See for example: https://doi.org/10.3389/fvets.2019.00154.

Answer: Thank you again for your great suggestion which has been incorporated, lines 736 – 739.

Line 514: Can the authors expand on an example of this if already exists in horses, or in other animals if published? The reference provided is not appropriate as it is not a computational approach (the landmarks are manually defined in each picture). A better example of what the authors are trying to say is perhaps this: https://doi.org/10.1016/j.conb.2019.10.008 (DeepLabCut Model Zoo has a module for primate faces with 45 markers that track movement).

Answer: Thanks for suggesting the reference, which has been included (lines 745 – 746).

Does FaceSed need training or specific backgrounds (e.g. only anaesthesiologists, only people very familiar with horses) to be used or anyone can use it? Please add that to your final remarks.

Answer: Included in lines 748 – 751.

The ears pictures all have zero above them.

Answer: Corrected

Answer: We really appreciate your comments.

Reviewer #2: dear authors,

you have performed a very nice study that will add to the knowledge on objective assessment of quality of sedation in horses. The reviewer only has some minor issues that should be addressed, among them there are some questions about the statistics you have used and about the way criteria validation has been performed:

Answer: Dear Reviewer

The authors appreciate your time and effort spent reviewing this manuscript, and thank you very much for your comments. All corrections have been performed according to the Reviewers' suggestions, and each comment responded to separately. Changes are highlighted according to the Journal's recommendations. We hope that after these corrections, you consider the manuscript suitable for publication, but we are happy to answer any further questions.

abstract:

line 23: "measuring" instead of "measured"

Answer: Corrected

line 25: "performed" instead of "perfomed"

Answer: Corrected

line 26: instead of "associated" you could better use the word "combined"

Answer: Changed to combined

line 32: you state that intra- and inter observer reliability were good to very good, but the reader does not know at this stage what the values 0.74-0.94 are? (ICC, Crohnbach's alpha??)

Answer: These analyses are ICC, thank you for your observation. The ICC has been included, line 32.

line 36: you state in this line that the scale was unidimensional according to principal component analysis, why? maybe some explanation over here?

Answer: The explanation has been included that all had load factors above 0.5 at the first dimension (line 38).

line 38: you state in this line that intem-total correlation was adequate, although the range of values is 0.3-0.73. It is not clear again what values these are (ICC?), but the lower range of 0.3 does not seem to be high enough for adequate?

Answer: The Spearman correlation has been included and so has the measurement in rho (line 40). This range follows the reference for item total correlation analysis that identify how much each item is correlated to the total score of the scale (Strainer et al., 2015). A detailed description about adequacy of this range is given in the discussion, lines 668-672.

introduction:

line 49: you mention standing horses, what about the role of sedative drugs in premedication for general anaesthesia?

Answer: This has been included in the 2nd sentence of the first paragraph of the introduction (lines 54-55).

line 64: instead of "sedation" you could maybe better use "depth and quality of sedation"?

Answer: The paragraph has been rephrased according to the suggestion of Reviewer 1, and depth and quality of sedation has been included throughout the manuscript where applicable.

materials and methods:

line 80-81 and line 95: what do you mean with "this study was opportunistic of another study"?

Answer: The word opportunistic has been replaced by other terms. The data of this study were collected during other parallel studies performed simultaneously and this has now been better explained in the manuscript (lines 126-129). 

line 90: instead of "has begun" you could better use "began" or "would begin"

Answer: Changed to began, line 157.

line 96-101: the reviewer thinks you should mention over here what the calculated sample size was?

Answer: The number of horses for the sample size calculation has been included. Line 165.

phase I:

line 107: instead of "data" you should use the word "parameter"

Answer: Changed to parameter, line 180.

line 113-114: what do you mean with "in situ registration"?

Answer: The sentence has been rephrased and in situ is no longer used. In situ has been replaced with on site throughout the manuscript

line 116: how was the treatment randomized?

Answer: Included in lines 193 – 195.

line 117: it is not clear what the website you mention over here should tell the reader?

Answer: The website has now been included as a reference according to the Journals instructions. It was used as a tool to randomize the treatments. Line 194.

line 118: "Those" probably reflects to the treatment, but this is not grammatically sound

Answer: “Those” has been changed to “the treatments” line 193.

Selection of photos and evaluations

line 151: "that" instead of "who"

Answer: Changed to “that”, line 238.

line 151: "plasma" instead of "plasmatic"

Answer: Changed to plasma, line 238.

line 151-152: what do you mean with "corroborated by the degree of sedation recorded in situ"??

Answer: The sentence has been rephrased to explain how and where the degree of sedation was evaluated. In situ has been changed to on site throughout the study (line 238 - 240).

line 161: isn't 120 minutes after ACP administration a bit early for the higher dosage of ACP to determine end of tranquilisation?

Answer: The final time point of 120 min was chosen for convenience, to avoid the restriction of the horses in the stall for longer periods. This has been included in lines 253. The experimental time-points chosen for Phase II have been further explained in lines 251- 252.

line 162: "did not communicate with each other" instead of "between them"

Answer: Included, line 256.

line 174-176: why did you choose this 80% as the correlation that was needed to start the main evaluations?

Answer: The Spearman correlation was performed before starting the main evaluations to guarantee that the evaluators were deeply involved in their observations and were, therefore, reliable. This has been included in lines 273 – 276. According to Streiner, 2015, the Interpretation of Spearman's correlation is 0 - 0.35 - low correlation; 0.35 - 0.7 - medium correlation; 0.7 - 1.0 - high correlation. We considered 80% a good starting point for the main evaluations. The reason why this correlation was chosen and the reference have been included in line 273 - 277.

line 183-184: you mention over here that the observers were asked to score the NRS first and after that the FaceSed. Why did you choose for this order? The main research question was about FaceSed and not about the NRS scoring, right? Might this previous NRS scoring have influenced the following FaceSed scoring? this could be a source of bias that is incorporated in the study design.

It would have ben much better if the NRS scores and FaceSed scores would have been disconnected from each other. Another problem is that you analyze the correlation between the NRS and the Face Sed, but because these were scored immediately after another, these observations were not independent and the correlation is therefore not valid.

Answer: Although the authors agree that it would be better to assess NRS disconnected from FaceSed, it is part of the validation process (specifically the concurrent criterion validation) to compare the proposed scale (FaceSed) against another one that is used for the same purpose (NRS) and at the same time. According to the literature mentioned in the manuscript this should be performed simultaneously (or sequentially). Considering the amount of data, it would not be feasible to perform one separate analysis for each scale. Other studies that validated pain scales used similar methodologies when working with extensive data sets. This topic has been included in the discussion as a bias, to consider the Reviewer´s point (lines 616 - 624).

line 193: "experience in sedation" instead of "experience of sedation

Answer: This sentence has been rephrased.

Statistical analyses

line 204: "...who attributed to the importance...."

Answer: Included, line 316.

line 210: what do you mean with "the following analysis"?

Answer: Changed to analysis described below, line 322.

line 211: what do you mean with in-situ?

Answer: In-situ has been changed to on site throughout the manuscript. 

line 230: for concurrent criteria validation, the reviewer thinks that it is not the most logical choice to compare the FaceSed with the NRS, since this scale also holds some subjectivity and since these were assessed and taken immediately after another, they aare not independent. Therefore, it would be better to compare FaceSed only with the HHAG%, since this is a completely objective and independent parameter.

Answer: The authors agree with the points raised by the reviewer about the NRS. This flaw has been included in the discussion as described above (lines 616 - 624). Both NRS and HHAG% were considered for concurrent criterion validity because as reported in lines 628 - 630, although HHAG% is a good method to assess depth of sedation, it is not very applicable for tranquilisation because asepromazine doses do not influence HHAG% in the same way as the alfa-2 agonists. Therefore, both HHAG% and NRS results are available for the reader. The paragraph about this topic has been rephrased and literature included to address the reviewer´s point and the reason behind the correlation between FaceSed and both instruments (NRS and HHAG%). 

line 241-242: this sentence is not grammatically sound, what do you want to explain with this sentence?

Answer: The paragraph has been rephrased (lines 360 – 361).

line 243: I think you would need to explain what the eigen values and variance mean. These are not necessarily parameters that all readers are familiair with.

Answer: Explanation included in lines 363 – 366. “The eigenvalues and variance are coefficients extracted from the correlation matrix of the PCA that indicate the degree of contribution of each dimension, helping to select only the representative dimensions”. 

line 255: the range of acceptable values for the item-total correlation of 0.3 to 0.7 seems to have a very low lower margin for acceptable Spearman correlation? A correlation of 0.3 would seem to be very low according and not acceptable to the reviewer?

Answer: The authors agree with the reviewer that correlations of 0.3 are low, but differently from other types of correlation calculation, the values of item-total correlation are interpreted differently. This range is supported by the validation procedures of the literature and the reason behind this range has now been better explained in the discussion (lines 668-672). Please let us know if this is now clear.

line 268: I would change the text into "...the frequency of sedation scores that were assigned by the evaluators...."

Answer: Changed (line 414).

line 272: you could add "statistical significance was accepted at P<0.05?

Answer: Included - line 422.

Results

line 295: the range of repeatability of the FaceSed sum of 0.74-0.94 seems to be the combined range of all 4 observers taken together. This should be mentioned like this in this line.

Answer: Included - line 445.

line 298: instead of "steps" it would be better to use the term "observations"

Answer: Corrected– line 450.

line 372: like mentioned earlier, a correlation of 0.3 does not seems to be acceptable according to the reviewer.

Answer: Please find the explanation above. This has been included in the discussion (lines 668 - 672).

line 373: in this line you state that all parameters showed an item-total correlation that was acceptable (although I would doubt this with a correlation of 0.3), except for the item eyes. What was the item-total correlation for this parameter?

Answer: Included between brackets (line 528) and the result is also presented in Table 6.

line 378-379: The Crohynbach's alpha was used as a measure of internal consistency. With what was the FaceSed score compared to determine this value?

Answer: The internal consistency assessed by Cronbach's alpha coefficient investigates whether the items of the scale are showing a consistent (similar) response. When a scale is developed, it is expected that the scores would correlate well mutually. Essentially, internal consistency represents the average of the correlations among the items of the scale. A more detailed explanation has been included in the methods (line 369) and discussion (lines 662 - 664).

Discussion:

line 409: "low degree of sedation" instead of "low sedation"

Answer: Corrected, line 575.

line 427-428: could you maybe discuss this difference, for instance due to the technically more difficult task of observing ridden horses?

Answer: Included, lines 601 – 603.

line 432: "the biases that could affect.."

Answer: Included, line 605.

line 436-437: you are right that scoring of the FaceSed did not bias scoring of the NRS, but the other way around, there might be a reason for bias. The fact that the NRS was first scored and immediately after that the FaceSed, might have influenced the reproducibility and repeatability of FaceSed and the correlation between FaceSed and NRS. This is what you mention in line 446, however, this high correlation might be due to how the scoring was performed.

Answer: Thanks for pointing that out. This bias has been included in lines 615-624 as well as the reason why they had to be simultaneously assessed for concurrent criterion validity (lines 622 - 624).

line 448: this is not how you describe it earlier in your materials and methods: there, you say that for criterion validity, the FaceSed is compared to the NRS and the HHAG%.

Answer: The paragraph has been rephrased. The HHAG% is considered the gold standard, however, both the HHAG% and the NRS were used for comparing to FaceSed as described in other studies. Lines 632-639.

line 461: instead of "...tranquilisation and low and high sedation.." you could better formulate "...tranquilisation with ACP from low and high sedation intensities with detomidine (with or without methadone)". Maybe could you also hypothesize about a possible reason for this?

Answer: Corrected as suggested and the possible reason justified in lines 650 - 652. 

line 484: in this paragraph you discuss sensitivity and specificity. Was it possible to determine vcut-off values that optimally discriminate between non-sedated and sedated horses?

Answer: The cut-off value has been included in the statistical analysis, lines 397 – 411 and the results 546 – 556.

line 494-497: you describe the difference betwen orbital closure due to being relaxed compared to due to being in pain. This lack of discriminatory power on this parameter could also lead to a false positive score, could you underline this better maybe?

Answer: The sentence has been rephrased and the suggestion of the reviewer included, lines 704 – 708.

line 513: instead of "of", you could better use "..tranquilisation from..."

Answer: corrected line 743.

---

## [Decision Letter · Decision Letter 1]

19 Apr 2021

PONE-D-20-32183R1

Development and validation of the facial scale (FaceSed) to evaluate sedation in horses

PLOS ONE

Dear Dr. Luna,

Thank you for submitting your manuscript to PLOS ONE. After careful consideration, we feel that it has merit but does not fully meet PLOS ONE’s publication criteria as it currently stands. Therefore, we invite you to submit a revised version of the manuscript that addresses the points raised during the review process.

We look forward to receiving your revised manuscript.

Kind regards,

Chang-Qing Gao

Academic Editor

PLOS ONE

Journal Requirements:

Reviewers' comments:

Reviewer's Responses to Questions

**Comments to the Author**

1. If the authors have adequately addressed your comments raised in a previous round of review and you feel that this manuscript is now acceptable for publication, you may indicate that here to bypass the “Comments to the Author” section, enter your conflict of interest statement in the “Confidential to Editor” section, and submit your "Accept" recommendation.

Reviewer #1: (No Response)

2. Is the manuscript technically sound, and do the data support the conclusions?

Reviewer #1: Yes

3. Has the statistical analysis been performed appropriately and rigorously? 

Reviewer #1: Yes

4. Have the authors made all data underlying the findings in their manuscript fully available?

Reviewer #1: Yes

5. Is the manuscript presented in an intelligible fashion and written in standard English?

Reviewer #1: Yes

6. Review Comments to the Author

Reviewer #1: I re-read this study with a lot of interest once again. I congratulate the authors in carefully responding to all the comments and making changes accordingly. I believe the manuscript is now much more informative and clearer. Although it is a long read, it is a good example of a study that has enough detail for replication and for fully understanding what has been done. I have just a couple of minor comments now, please see below. And also a small note about line numbering in the highlighted document with the changes. I know this is a painstakingly process to get right (I always struggle myself whenever doing it in my own papers), but most of the lines mentioned in the answer are not the same as the highlighted manuscript. This made me take a little while longer to find all the changes, so it is something to keep in mind on future reviews. In any case, well done!

---Minor comments:---

Line 43-45: Explain briefly why is this a limitation. Is it one limitation or two? As in, using pictures AND horses were docile? (This might be explained in the text, but I was a bit confused when reading it here only)

Line 75: Do these scales require experience with the scale and/or horses? So only certain expert observers can use them? Is there any assessment of quality for how the observers use these scales (other than inter-observer reliability)?

While the clarifications and the author answers were all fantastic, I still struggled to follow the protocols and methodologies, particularly with the different phases and what belongs to what study. What about having a flow diagram to make it clear and easy to understand what was done, when, and for what studies?

7. PLOS authors have the option to publish the peer review history of their article (what does this mean?). If published, this will include your full peer review and any attached files.

Reviewer #1: No

---

## [Author Response · Author response to Decision Letter 1]

28 Apr 2021

PONE-D-20-32183

Development and validation of the facial scale (Facesed) to evaluate sedation in horses

Reviewer #1: I re-read this study with a lot of interest once again. I congratulate the authors in carefully responding to all the comments and making changes accordingly. I believe the manuscript is now much more informative and clearer. Although it is a long read, it is a good example of a study that has enough detail for replication and for fully understanding what has been done. I have just a couple of minor comments now, please see below. And also a small note about line numbering in the highlighted document with the changes. I know this is a painstakingly process to get right (I always struggle myself whenever doing it in my own papers), but most of the lines mentioned in the answer are not the same as the highlighted manuscript. This made me take a little while longer to find all the changes, so it is something to keep in mind on future reviews. In any case, well done!

Answer: 

Dear Reviewer

We appreciate your comments and thank you for your final suggestions.

Please find below each point answered separately.

Line 43-45: Explain briefly why is this a limitation. Is it one limitation or two? As in, using pictures AND horses were docile? (This might be explained in the text, but I was a bit confused when reading it here only)

Answer: included

Line 75: Do these scales require experience with the scale and/or horses? So only certain expert observers can use them? Is there any assessment of quality for how the observers use these scales (other than inter-observer reliability)?

Answer: These scales will require experience “with the effects of sedation in horses” (included). This point has also been addressed in the last sentence of the same paragraph “Unidimensional scales (VAS, NRS and SDS) may be biased to the interpretation and experience of the evaluator, generating differences in results with doubtful representativeness when comparing studies”. 

While the clarifications and the author answers were all fantastic, I still struggled to follow the protocols and methodologies, particularly with the different phases and what belongs to what study. What about having a flow diagram to make it clear and easy to understand what was done, when, and for what studies?

Answer: a flow chart has been included as requested.

---

## [Decision Letter · Decision Letter 2]

6 May 2021

Development and validation of the facial scale (FaceSed) to evaluate sedation in horses

PONE-D-20-32183R2

Dear Dr. Luna,

We’re pleased to inform you that your manuscript has been judged scientifically suitable for publication and will be formally accepted for publication once it meets all outstanding technical requirements.

Kind regards,

Chang-Qing Gao

Academic Editor

PLOS ONE

Additional Editor Comments (optional):

Reviewers' comments:

Reviewer's Responses to Questions

**Comments to the Author**

1. If the authors have adequately addressed your comments raised in a previous round of review and you feel that this manuscript is now acceptable for publication, you may indicate that here to bypass the “Comments to the Author” section, enter your conflict of interest statement in the “Confidential to Editor” section, and submit your "Accept" recommendation.

Reviewer #1: All comments have been addressed

2. Is the manuscript technically sound, and do the data support the conclusions?

Reviewer #1: Yes

3. Has the statistical analysis been performed appropriately and rigorously? 

Reviewer #1: Yes

4. Have the authors made all data underlying the findings in their manuscript fully available?

Reviewer #1: Yes

5. Is the manuscript presented in an intelligible fashion and written in standard English?

Reviewer #1: Yes

6. Review Comments to the Author

Reviewer #1: Thanks for addressing the few last comments. Great flow chart as well, which now makes it really clear what was done. I have no further comments.

7. PLOS authors have the option to publish the peer review history of their article (what does this mean?). If published, this will include your full peer review and any attached files.

Reviewer #1: No

---

## [Editor Report · Acceptance letter]

20 May 2021

PONE-D-20-32183R2 

Development and validation of the facial scale (FaceSed) to evaluate sedation in horses. 

Dear Dr. Luna:

I'm pleased to inform you that your manuscript has been deemed suitable for publication in PLOS ONE. Congratulations! Your manuscript is now with our production department. 

Kind regards, 

on behalf of

Dr. Chang-Qing Gao 

Academic Editor

PLOS ONE